# The Mediterranean climate change hotspot in the CMIP5 and CMIP6 projections

Josep Cos[1], Francisco Doblas-Reyes[1,2], Martin Jury[1,3], Raül Marcos[1], Pierre-Antoine Bretonnière[1], and Margarida Samsó[1]

[1]Earth Sciences Department, Barcelona Supercomputing Center (BSC), Barcelona, Spain.
[2]Institució Catalana de Recerca i Estudis Avançats (ICREA), Barcelona, Spain
[3]Wegener Center for Climate and Global Change, University of Graz, Graz, Austria

**Correspondence:** Josep Cos (josep.cos@bsc.es)

**Abstract.** The enhanced warming trend and precipitation decline in the Mediterranean region make it a climate change hotspot. We compare projections of multiple CMIP5 and CMIP6 historical and future scenario simulations to quantify the impacts of the already changing climate in the region. In particular, we investigate changes in temperature and precipitation during the 21st century following scenarios RCP2.6, RCP4.5 and RCP8.5 for CMIP5 and SSP1-2.6, SSP2-4.5 and SSP5-8.5 from CMIP6, as well as the HighResMIP high resolution experiments. A model weighting scheme is applied to obtain constrained estimates of projected changes, which accounts for historical model performance and inter-independence of the multi-model ensembles, using an observational ensemble as reference. Results indicate a robust and significant warming over the Mediterranean region during the 21st century over all seasons, ensembles and experiments. The temperature changes vary between CMIPs, CMIP6 being the ensemble that projects a stronger warming. The Mediterranean amplified warming with respect to the global mean is mainly found during summer. The projected Mediterranean warming during the summer season can span from 1.83 to 8.49 ºC in CMIP6 and 1.22 to 6.63 ºC in CMIP5 considering three different scenarios and the 50% of inter-model spread by the end of the century. Contrarily to temperature projections, precipitation changes show greater uncertainties and spatial heterogeneity. However, a robust and significant precipitation decline is projected over large parts of the region during summer by the end of the century and for the high emission scenario (-49 to -16 % in CMIP6 and -47 to -22 % in CMIP5). While there is less disagreement in projected precipitation between CMIP5 and CMIP6, the latter shows larger precipitation declines in some regions. Results obtained from the model weighting scheme indicate larger warming trends in CMIP5 and a weaker warming trend in CMIP6, thereby reducing the difference between the multi-model ensemble means from 1.32 ºC before weighting to 0.68 ºC after weighting.

## 1   Introduction

The Mediterranean region (10º W, 40º E, 30º N, 45º N) (Iturbide et al., 2020) is located between the arid and warm North-African climate and the humid and mild European climate (Cramer et al., 2018). The contrast between them is partly explained by the influence of the surrounding oceans, their interaction with the land surface and the general atmospheric circulation characteristics in mid-latitudes (Boé and Terray, 2014).

Global warming is not homogeneous, and Lionello and Scarascia (2018) suggests that the Mediterranean region is a climate change hotspot. Consequently, adaptation to the changing climate threats is paramount to the countries located around the Mediterranean Sea (Gleick, 2014; Cramer et al., 2018), which live in a complex and diverse socioeconomic situation and have severe vulnerabilities to climate change and variability (Barros et al., 2014). The observed warming in the Mediterranean region during the last decades is expected to continue and grow larger than the global-mean warming (Lionello and Scarascia, 2018). Additionally, total precipitation declines have been observed during the late 20th century (Longobardi and Villani, 2010), and are projected by different multi-model ensembles for the 21st century (Paeth et al., 2017; Zittis et al., 2019). Characteristics of the projected Mediterranean climate change have been linked to thermodynamic sources such as land-ocean warming contrast and lapse rate change in summer (Brogli et al., 2019), or to dynamical processes such as the changes in upper-tropospheric large-scale flow in winter (Tuel and Eltahir, 2020).

Numerical models are used to estimate future climate change. Accounting for the physical processes and interactions in each climate subsystem (atmosphere, biosphere, cryosphere, hydrosphere and land-surface), global climate models (GCMs) aim to project the state of the future climate system. Model runs over long historical or future periods are driven by natural forcings (i.e. solar irradiance and volcanic aerosols) and anthropogenic emissions that alter the greenhouse gas (GHG) concentrations, leading to changes in the radiative forcing. (Hawkins and Sutton, 2011). GCMs are developed by a number of institutions who always apply the same physical principles but might use slightly different assumptions. This opens the door to performing the same experiments with multiple GCM outputs leading to more robust estimates. Modelling uncertainty can be sampled by ensembling various models (Tebaldi and Knutti, 2007), while running the same model multiple times (referred to as members) under the same experiment samples internal variability (Hawkins and Sutton, 2011). To make the results comparable, intercomparison projects where several models perform standardized experiments have been organised by the international community. (Meinshausen et al., 2011; Riahi et al., 2016). The main community effort is the Coupled Model Intercomparison Project (CMIP). In this study we consider the latest two CMIP phases, CMIP5 and CMIP6 (Taylor et al., 2012; Eyring et al., 2016), and explore their similarities and differences over the Mediterranean region. The almost ten years between CMIP5 and CMIP6 allowed for improvements in the modelling of certain earth system processes such as cloud feedbacks, aerosol forcings or aerosol-cloud interactions (Voosen, 2019; Wang et al., 2021).

CMIP experiments were performed with a large set of models and therefore show many differences in projected changes due to internal variability and the diverse model designs used by the modelling teams. By weighting single model runs according to their performance in simulating the observed past allows constraining the climate modelling uncertainty and obtaining a potentially more accurate estimate of regional climate change signals. Various studies have used different subsetting/weighting approaches such as emergent constraints (Cox et al., 2018; Hall et al., 2019; Tokarska et al., 2020), performance-based model subsets (McSweeney et al., 2015; Langenbrunner and Neelin, 2017; Herger et al., 2019) or model weighting accounting for performance and independence (Knutti et al., 2017; Lorenz et al., 2018; Brunner et al., 2019). The latter approach has been used in this study as it additionally considers the interdependencies existing between the models.

This study evaluates and quantifies the Mediterranean climate change hotspot for each season over the 21st century by looking into surface air temperature and precipitation changes in the Mediterranean and how they relate to larger-scale responses.

We consider three different emission scenarios to assess the impact of anthropogenic emission uncertainties over the Mediterranean climate. The CMIP5 and CMIP6 multi-model ensembles are used to estimate the climate change signal, its uncertainty and to illustrate the differences between the two experiments in the region. Finally, a weighting method is applied to each CMIP ensemble based on the criteria of model performance and independence to obtain more robust projections.

Section 2 describes the climate models and observational data used, and explains the methods to quantify climate change and weight the projection members. The climate change hotspot in the Mediterranean and the weighted and unweighted projected changes are presented in section 3, while these results are discussed in section 4. Section 5 concludes and raises questions for further investigation.

## 2 Data & Methods

### 2.1 Model data

This study is based on the CMIP5 and CMIP6 historical and future climate projections experiments. The historical CMIP5 experiments span from 1850 to 2005 (Taylor et al., 2012) and from 1850 to 2014 in CMIP6 (Eyring et al., 2016). The future projections are a continuation of the historical simulations, and we have used runs spanning until year 2100. The variables are monthly mean near-surface air temperature (TAS), precipitation rate (PR) and sea-level pressure (PSL). The latter is used to weight the ensemble members together with TAS (see section 2.3).

The increasing computational power over time has allowed for increased model resolution and complexity, which leads to the expectation that models have improved from CMIP5 to CMIP6. Additionally, we have used the High Resolution Model Intercomparison Project (HighResMIP), a CMIP6 endorsed MIP (Haarsma et al., 2016), aiming to compare lower and higher resolution versions of the same global models. The historical and future HighResMIP periods span from 1950 to 2014 and 2015 to 2050, respectively. Though only a subset of the CMIP6 models contributed to HighResMIP, this smaller ensemble has been also considered in this study to assess the impact of increasing model resolution on the Mediterranean climate.

Three radiative forcing scenarios are used to account for uncertainty in future emissions: the CMIP5 Representative Concentration Pathways (RCPs; (van Vuuren et al., 2011)) 2.6, 4.5 and 8.5 and the CMIP6 Shared Socioeconomic Pathways (SSPs; (Riahi et al., 2016)) 1-2.6, 2-4.5 and 5-8.5. The magnitudes 2.6, 4.5 and 8.5 (in $Wm^{-2}$) represent the 2100 global radiative forcing in comparison to the pre-industrial era. However, even if the radiative forcing at the end of the century is the same in both RCPs and SSPs, the path to reach it can differ substantially, leading to differences in the projected climate (Wyser et al., 2020). One of the main differences between the SSPs and RCPs is that the former have a compatible socioeconomic scenario associated to each forcing scenario, SSP1 being based on sustainability, inclusive development and inequality reduction, SSP2 representing a middle of the road scenario, where slow progress is made in achieving sustainable development goals and with a mild decline in resource and energy use, and SSP5 based on a fossil-fueled development, rapid technological progress and economic growth (Riahi et al., 2016; O'Neill et al., 2016). The results from CMIP5 and CMIP6 sharing the same 2100 radiative forcing will be displayed together for simplicity, but the reader should always bear in mind that the evolution of GHG concentrations differs between them. They are not entirely comparable as RCPs and SSPs defined with the same radiative

forcing at the end of the century do not share the same progression of aerosol and GHG concentrations along the 21st century. HighResMIP is only available for the scenario SSP5-8.5 for future projections.

Many of the models have more than one member, meaning that the model runs have been started with different initial conditions leading to diverging climate trajectories. The aim of having multiple members is to sample the uncertainty that arises from internal variability (Lehner et al., 2020; Deser et al., 2020). Having multi-member models means that the multi-model ensembles are super-ensembles. A summary of the simulations performed by each model used and for every scenario can be found in Appendix A.

## 2.2 Observational data

We use observational references to compare the model experiments to the observed past and to derive performance weights of ensemble members. Multiple observational products are used including both reanalysis (ERA5 and JRA55) and gridded observations (GPCC, CRU, BerkeleyEarth and HadSLP2) to account for observational uncertainty. A summary of the observational datasets used is found in Table 1. JRA55 will not be displayed in the time series plots as it overestimates the precipitation over the Mediterranean during the period 1958-1978 (Tsujino et al., 2018).

**Table 1.** Summary of the observational references for near-surface air temperature (TAS), precipitation rate (PR) and sea-level pressure (PSL).

| Name | Type | Institute | Variables | Reference |
|------|------|-----------|-----------|-----------|
| JRA55 | Reanalysis | Japan Meteorological Agency (JMA) | TAS, PR, PSL | (Kobayashi et al., 2015) |
| ERA5 | Reanalysis | European Centre for Medium-Range Weather Forecasts (ECMWF) | TAS, PSL | (Hersbach et al., 2020) |
| CRU (v4.04) | Gridded observations | University of East Anglia (UEA) | TAS, PR | (Harris et al., 2020) |
| GPCC (v2018) | Gridded observations | Deutscher Wetterdienst (DWD) | PR | (Schamm et al., 2014) |
| BerkeleyEarth | Gridded observations | Berkeley Earth | TAS | (Rohde et al., 2013) |
| HadSLP2 | Gridded observations | Met Office (UKMO) | PSL | (Allan and Ansell, 2006) |

## 2.3 Methods

All datasets are regridded to a $1^o \times 1^o$ grid using a conservative interpolation method to allow the comparison between different models and observational references. After regridding, the dataset's original orography will differ from that of the $1^o \times 1^o$ grid. Therefore, the TAS values obtained for a specific altitude might suffer a shift in altitude which needs to be corrected by means of the 6.49 $K/km$ standard lapse rate (Weedon et al., 2011; Dennis, 2014). This is only necessary when absolute climatologies are used, as computing the change in TAS climatology from one period to the other cancels out this height shift.

To assess the seasonal dependence of climate change over the Mediterranean region, results are computed for December-January-February (DJF), March-April-May (MAM), June-July-August (JJA) and September-October-November (SON). A summary of the time periods used and and the applications of the different diagnostics can be found in Table 2.

All calculations have been performed using the Earth System Model Evaluation Tool (ESMValTool). ESMValTool is a
community framework that facilitates the processing of generic climate datasets, allowing for reproducibility of results (Righi
et al., 2020).

Mediterranean TAS and PR are assessed over land to highlight the impact of climate change over populated regions. This
avoids values over sea influencing results over land when the regridding is performed, i.e. TAS behaves differently over land
than over sea due to differences in surface thermodynamic properties, while PR over sea should not have an impact on fresh-
water resources over land.

### 2.3.1 Projections verification

To verify the projection ensembles used, we compare the linear trend (TREND) distributions of the observational products
against the multi-model ensembles. It is computed by applying the linear ordinary least square regression fit with time as
an independent variable. The 35-year period 1980-2014 has been used to calculate each of the model's and observational
dataset's trends, as a period with shorter span would be too dependant on the effect of internal variability from the climate
system (Merrifield et al., 2020; Peña-Angulo et al., 2020). Note that CMIP5 years 2006-2014 are taken from the corresponding
scenario simulation. The results are gathered in the respective OBS, CMIP5, CMIP6 and HighResMIP distributions (displayed
as box plots), and we perform a qualitative assessment on the differences between observed and simulated historical trends.

### 2.3.2 Mediterranean hotspot evaluation

A climate change hotspot is defined as a region whose climate is especially responsive to global change (Giorgi, 2006). To
characterize the hotspot, we compare the TAS and PR behaviours in the Mediterranean against the global and latitudinal band
responses, respectively. The first step is to calculate the change in the variables' magnitude between the reference period [1986-
2005, from Collins et al. (2013)] climatology (CLIM) and a future period CLIM (this diagnostic is referred to as $\Delta$ in this text).
To evaluate the TAS hotspot we compute the differences between each model's Mediterranean land-only $\Delta$TAS and the global
land-ocean $\Delta$TAS mean. For PR the land-ocean latitudinal belt 30º N-45º N mean is used instead of the global mean (Lionello
and Scarascia, 2018). Once the whole ensemble differences are calculated the multi-model mean is computed.

To highlight the difference in the impact of the hotspot within the Mediterranean region we plot the hotspot maps using the
near-term and long-term $\Delta$, which refer to the future periods 2041-2060 and 2081-2100, respectively. Additionally, To assess
the evolution of the hotspot we calculate the projected area-averaged 10-year rolling windows of the Mediterranean $\Delta$ and the
large-scale $\Delta$ for both TAS and PR. For precipitation, the area aggregations are computed using absolute values and then the
relative change with respect to the reference is calculated (displayed in %).

### 2.3.3 Mediterranean projected changes quantification

To quantify the projected magnitudes of the Mediterranean region climate change we compute the $\Delta$ between the reference
period 1985-2005 and the future periods: near-term (2021-2040), mid-term (2041-2060) and long-term (2081-2100). We use

20-year baseline and future periods following the guidelines from IPCC (2021). Additionally, as CMIP5 historical simulations end in 2005, the reference period 1986-2005 from IPCC's AR5 (Collins et al., 2013) is chosen to avoid overlapping historical and scenario experiments when extracting projection results. Note that only the near-term period is available for HighResMIP as the future experiment ends in 2050. The advantage of using $\Delta$ instead of future CLIMs is that GCMs mean-state systematic biases are removed, and we obtain a more easily interpretable comparison of the responses among models and between models

and observations (Garfinkel et al., 2020).

     With the aim to sample the inherent uncertainty of the multi-model ensemble, we compute the inter-model spread from the 5th and 95th percentiles of the ensemble distribution. To take into account the scenario uncertainty we display side by side the distribution of $\Delta$ from the three different scenarios that we have used for each ensemble (RCP2.6, RCP4.5 and RCP8.5 for CMIP5 and SSP1-2.6, SSP2-4.5 and SSP5-8.5 for CMIP6)

The statistical significance of TAS and PR mean changes and the degree of agreement between models are used to assess the uncertainty and robustness of the multi-model ensemble results. A climate change signal is considered robust when at least 80 % of the models agree on the projected sign of the $\Delta$s (Collins et al., 2013). A change in the multi-model mean is considered significant when it is beyond the threshold of a two-tailed t-Student test at the 95 % confidence level. The historical and future ensemble mean change and their inter-model standard deviations are used to compute the t-statistic.

### 2.3.4    Weighting method

     It has been argued that more robust projections could be obtained by giving more weight to members with good performance (Knutti et al., 2017). Therefore, we compare historical simulations against the observational ensemble mean and more weight is given to those members that better reproduce the observed climate i.e. weighting them by performance. Another aspect that can be taken into account when weighting a multi-model ensemble is the independence between members. Giving equal weight to all members (one model one vote) is not a fair approach as some share model formulations (either because their runs belong

to the same model or because their models share similarities), and would be overrepresented in the ensemble. An independence weighting method is applied to correct this issue.

     Using the approach developed in Lorenz et al. (2018), Brunner et al. (2020) and Merrifield et al. (2020), we use equation (1) to give a weight $w_i$ to each member $i$ in the projections ensemble. The distances (measured with the root mean squared

error, RMSE) $D_i$ between member $i$ and the observational reference inform the performance weight, and the distance $S_{ij}$ between member $i$ and every other member $j$ from the multi-model ensemble informs the independence weight. The amount of $j$ members is represented by $m$, which is the total number of members minus one. $\sigma_s$ and $\sigma_d$ are the independence and performance shape parameters, respectively. The mean of the observational ensemble is used as the observational reference.

$$w_i = \frac{e^{-\left(\frac{D_i}{\sigma_d}\right)^2}}{1 + \sum_{j \neq i}^{m} e^{-\left(\frac{S_{ij}}{\sigma_s}\right)^2}} \tag{1}$$

The weighting method distances account for different performance and independence diagnostics (trends, differences, variabilities and climatologies) to avoid weighting members that could match the performance and independence criteria of a single

diagnostic just by chance. The diagnostics $d_i$ and $s_{ij}$ used to evaluate the distances $D_i$ and $S_{ij}$, respectively, are different as Merrifield et al. (2020) suggests. The aim when evaluating performance is to give more weight to members that resemble the observed past in a more faithful way. Differently, the aim of weighting for independence is to clearly identify members that behave in a similar way. All the diagnostics are computed over the period 1980-2014 (Brunner et al., 2020). The variables used to compute the diagnostics are TAS and PSL (Merrifield et al., 2020). The performance diagnostics are the surface temperature 1980-2014 CLIM minus its area average (TAS-DIFF), the surface temperature interannual standard deviation (TAS-STD); the surface temperature linear trend (TAS-TREND), the sea-level pressure 1980-2014 CLIM minus its area-average (PSL-DIFF), and the sea-level pressure interannual standard deviation (PSL-STD). The independence diagnostics are the 1980-2014 PSL and TAS climatologies (PSL-CLIM and TAS-CLIM).

The distances between member-observations for each of the diagnostics are aggregated as in equation 2 where $d_i$ represents the distance for each diagnostic $X^d = (\text{TAS-TREND}, \text{TAS-DIFF}, \text{TAS-STD}, \text{PSL-DIFF}, \text{PSL-STD})$. Equation 3 shows how to compute the distances between models and observations, where $g$ refers to each grid cell and $w_g$ represents its area weight. To find $S_{ij}$ the same method is followed but using $X^s = (\text{TAS-CLIM}, \text{PSL-CLIM})$ and comparing members against each other instead of observations.

$$D_i = \sum_{X^d} \frac{d_i^{X^d}}{\text{MEDIAN}_i(d_i^{X^d})} \qquad (2)$$

$$d_i^{X^d} = \sqrt{\sum_g w_g (X_i^d - X_{obs}^d)^2} \qquad (3)$$

The shape parameters are constant values that inform if the member-observations or the member-member distances are enough to downweight a member ($\sigma_d$) or if they are close enough to determine some dependency between members ($\sigma_s$), respectively. Each ensemble (CMIP5 and CMIP6), season and scenario has its own shape parameters associated. Appendix B explains in further detail the meaning of the shape parameters, the methods used to compute them and the diagnostics to determine performance and independence.

**Table 2.** Summary of each diagnostic's use and time period.

| Diagnostic | Period/s | Use |
| --- | --- | --- |
| $\Delta$ | 2021-2040/2061-2080/2081-2100 against 1986-2005 | weighted and unweighted projection results |
| DIFF | 1980-2014 | performance weight |
| STD | 1980-2014 | performance weight |
| TREND | 1980-2014 | performance weight and verification |
| CLIM | 1980-2014 | independence weighting |

## 3  Results

Apart from the figures displayed in this section and the supplementary material, additional ones generated during the study can be found in a shiny app in the following link https://earth.bsc.es/shiny/medprojections-shiny_app/.

### 3.1  Verification

We compare CMIP and HighResMIP ensemble TAS and PR trends to the observational ensemble trends between 1980 and 2014 as an indication of model performance over the Mediterranean. The spread of the multi-model ensemble trends contain the observational ensemble trends (see Fig. 1). Mostly, for seasons SON and MAM, the observations fall inside the 90 % spread of the multi-model ensemble historical runs (not shown). The historical multi-model ensemble spread of temperature trends is notably larger than that of the observational ensemble. CMIP6 past warming trends are generally larger than CMIP5. The inter-model spread for the precipitation projections is large for all ensembles and usually has both negative and positive trends (e.g DJF CMIP5 precipitation trends range from -0.092 to 0.097 $mm\,day^{-1}\,decade^{-1}$ for the 5th and 95th percentiles, respectively). HighResMIP TAS trends are contained within the CMIP6 ensemble, but some of the high-resolution (HR) models exhibit trends outside the CMIP6 range for PR in summer (Fig. 1.d). The agreement between the different observational products in past warming trends is shown in Fig. S7 (columns 1 and 5). While the general warming patterns are similar there are some notable differences over the Balkans and Western Asia. The figure also highlights the need of considering multiple observational sources, as historical trends differ both in magnitudes and spatial patterns.

### 3.2  The Mediterranean as a climate change hotspot

Figure 2 shows CMIP5 and CMIP6 high radiative forcing scenario differences of ΔTAS over the Mediterranean against the 1986-2005 global mean ΔTAS (for winter, summer and the annual means). The Mediterranean ΔPR is compared to the 30º N-45º N latitudinal belt ΔPR mean.

The Mediterranean region shows a higher annual temperature increase than the global mean. When accounting for seasonal differences, the highest amplifications are visible for summer over the Iberian Peninsula and the Balkans. CMIP5 and CMIP6 agree on the regions showing the highest amplified warming, but the latter projects larger amplification magnitudes. There is agreement between both CMIPs in the distribution and magnitude of the winter warming amplification, which is small and even negative in the northwest part of the domain. While projections agree on a precipitation increase in the 30º N-45º N latitudinal belt for the long-term period (Lionello and Scarascia, 2018), the Mediterranean region shows a decline in precipitation. The largest amplified drying shifts latitudinally from the south of the Mediterranean region in winter to the north in summer. The most affected region in summer is projected to be the southwest of the Iberian Peninsula. Both CMIPs agree on the precipitation patterns of change, but CMIP6 dries more and faster in the amplified drying regions, and projects larger precipitation increases in regions where the hotspot has a negative sign such as the southeast of the domain (probably enhanced by using relative precipitation changes).

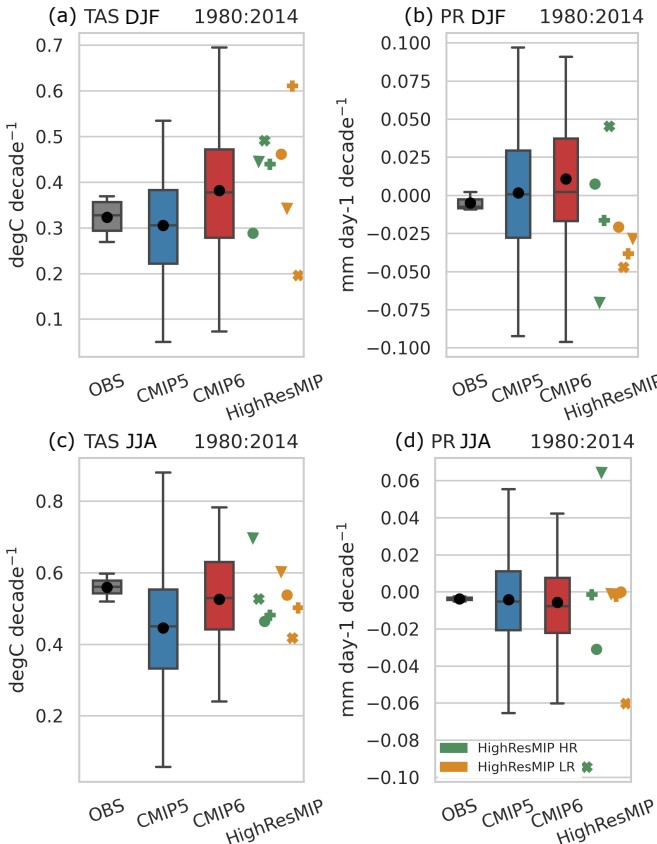

**Figure 1.** Historical trends for winter (a, b) and summer (c, d) temperature (a, c) and precipitation (b, d) of the observational, CMIP5, CMIP6 and HighResMIP ensembles. The observational distribution is composed of the different values obtained from each of the observational products. In the box plots, the black horizontal line represents the median and the black dot is the mean. The interquartilic range (IQR) and whiskers are defined by the 25th-75th and 5th-95th percentiles, respectively. HighResMIP models are displayed as markers, enabling a comparison of the HR (green) and LR (orange) models within the experiment. The same markers are used for two different resolution runs of the same model (see Table S1)

TAS and PR differences increase in magnitude from the mid to the long-term, while the spatial pattern remains the same, indicating that the climate in the Mediterranean changes faster than the global average when forced by the 8.5 $Wm^{-2}$ scenarios. The low emission scenario, instead, shows a hotspot weakening from the mid to the long-term as the warming amplification is reduced and the precipitation differences are maintained (see Fig. S1). The weakening of the hotspot under the low emission scenario will be further explored below.

Even though CMIP6 is projecting a larger warming and drying amplification than CMIP5, Fig. 3 shows that CMIP5 and CMIP6 agree on the relation between global and local warming (slopes painted in the figures). This indicates that CMIP6 is not enhancing the hotspot with respect to CMIP5, but rather the higher amplified warming in the Mediterranean is the

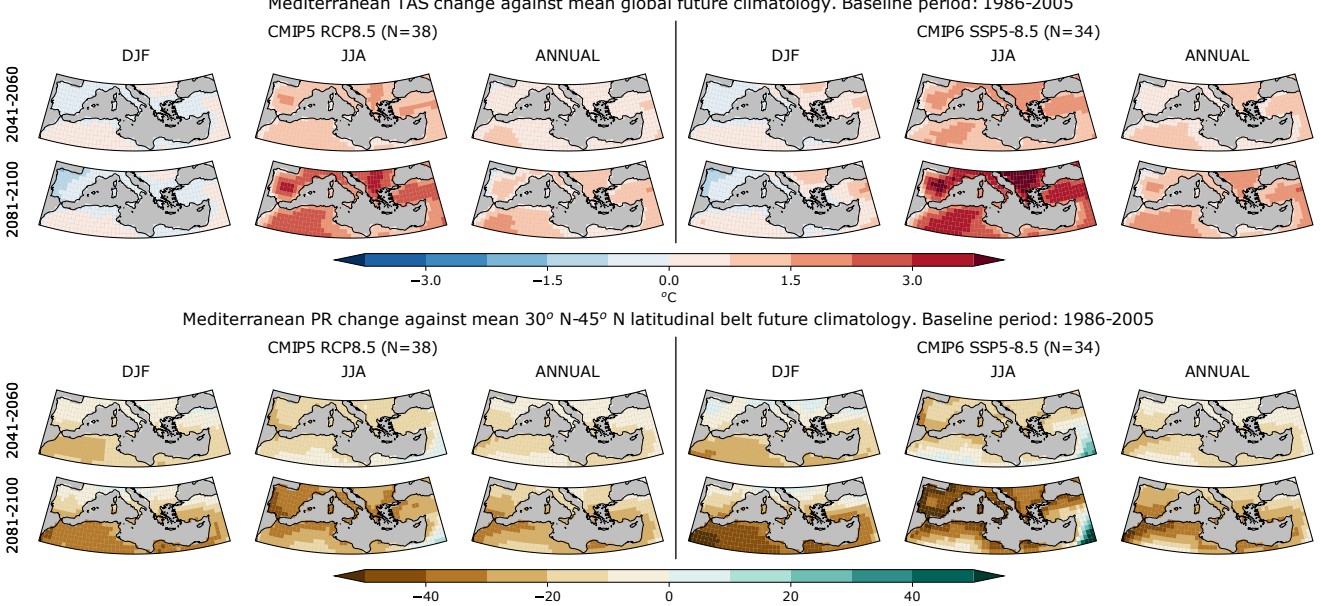

**Figure 2.** Mediterranean region TAS (upper rows) and PR (lower rows) change differences with respect to the mean global temperature change and the mean 30º N-45º N latitudinal belt precipitation change, respectively. The changes for periods 2041-2060 (1st and 3rd row) and 2081-2100 (2nd and 4th row) are evaluated against the 1986-2005 mean. The differences are shown for the CMIP5 (left) and CMIP6 (right) winter, summer and annual mean projections (columns) under the high emission scenario RCP8.5 and SSP5-8.5, respectively. N indicates the number of models included in the ensemble mean.

result of a globally warmer multi-model ensemble. For DJF, additional warming over the Mediterranean is almost zero with respect to the global mean. Contrastingly for JJA, additional warming over the Mediterranean is about $1.6 \times$ higher than the
240 global-mean warming. This relationship appears to be linearly maintained for higher global warming levels, i.e. with time and GHG-concentrations.

In spite of this strong agreement in the relationship between global and local warming, CMIP5 and CMIP6 have slight differences in the projected precipitation over the Mediterranean in comparison to the 30º N-45º N latitudinal belt (see Fig. S2). CMIP5 generally shows more negative slopes than CMIP6, meaning that the former is projecting a larger amplification of
245 the precipitation hotspot: as the relative precipitation loss in the Mediterranean (ordinate) for the same amount of precipitation increase in the larger scale region (abscissa) is larger. While this is true for all seasons and scenarios, the difference between CMIP5 and CMIP6 is more noticeable during winter and especially for the low emission scenario. Fig. S3 highlights more extreme CMIP6 precipitation relative changes in the latitudinal band and increases of over 30 % in Asia and over the Pacific as opposed to CMIP5. Therefore, conclusions must be drawn carefully from comparing area-averaged values of these regions.
Nevertheless, there is agreement between both ensembles on the spatial distribution of PR changes.

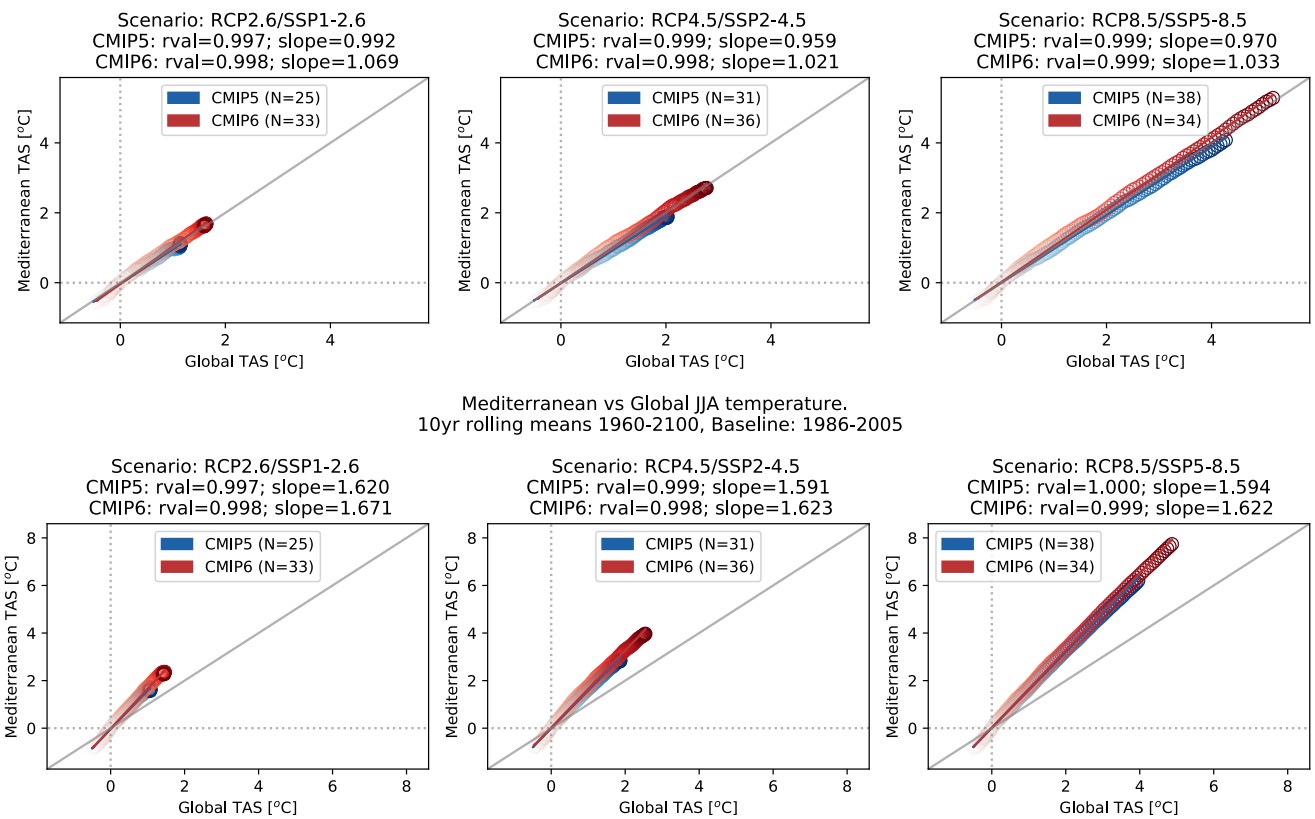

**Figure 3.** Mediterranean region warming against global warming for the three scenarios (columns) shown in winter (top row) and summer (bottom row) for the CMIP5 and CMIP6 ensemble means. Each dot represents a 10 year mean change beginning from 1960-1969 (light coloring) until 2091-2100 (opaque coloring). The changes are computed with 1986-2005 as baseline. An ordinary least squares linear regression is computed and the slope and $r$ values are shown. N indicates the number of models included in the ensemble mean.

We tried following a second approach to assess the trend differences of the precipitation hotspot between the CMIPs. Fig. S4 shows changes in precipitation for the Mediterranean region against the global mean warming and the ensemble that dries faster for the same magnitude of global warming is CMIP5. This is more noticeable during the winter season. The results from this figure, together with Fig. S2, gives some evidence on the fact that CMIP5 projects a larger precipitation hotspot (relative to its own large-scale climate response) than CMIP6.

Coming back to the hotspot weakening, the low emission scenario panels show more clearly how a recovery of the precipitation decline is projected following mitigation. For the rest of the scenarios, the projected amplified warming, combined with an anomalous precipitation decline, makes the Mediterranean a climate change hotspot (Lionello and Scarascia, 2018).

### 3.3 Unweighted projections

#### 3.3.1 Temperature

Figure 4(a) shows projected multi-model ensemble summer and winter TAS changes under three scenarios and three time horizons over the Mediterranean. The CMIP6 ensemble always shows larger $\Delta$TAS than CMIP5. Inter-model spread for the end of the century is larger for CMIP6 than CMIP5. CMIP6 projects summer temperatures to increase by over 7.4 ℃ (90 % inter-model spread within 5.6 ℃ to 9.1 ℃) until the end of the century under the high emission scenario and 2.3 ℃ (90 % within 1.2 ℃ to 3.3 ℃) under the low emission scenario (Fig. 4). CMIP5 shows a mean summer warming of 5.9 ℃ by the end of the century (90 % within 4.1 ℃ to 7.7 ℃) under RCP8.5 and 1.6 ℃ (90 % within 0.3 ℃ to 2.5 ℃) under RCP2.6. In winter the warming is always lower, and 90 % of CMIP6 models for the high emission scenario project a $\Delta$TAS between 3.3 and 6.8 ℃ (CMIP5: 2.7 ℃ to 5.0 ℃). For the remaining seasons (MAM and SON), CMIP6 shows a larger warming and larger intermodel spread than CMIP5 (not shown). HighResMIP HR and low-resolution (LR) projections are contained within the CMIP5 and CMIP6 distributions (only near-term, see Fig. S5.c). No specific relation between the LR and HR model outputs can be find, and due to the small size of the HighResMIP ensemble further conclusions cannot be drawn. Finally, from the area averaged distributions of $\Delta$TAS (Fig. 4.a) we can see that the largest source of uncertainty for the mid and long-terms is the forcing scenario, and the inter-model spread for the near-term.

The inter-model spread grows larger with emissions both for TAS and PR (Fig. 4(a,c)). To check the influence of the equilibrium climate sensitivity (ECS) on the increasing inter-model spread, the same plot is computed with a subset of CMIP5 and CMIP6 models with ECSs constrained between 2.6 and 3.3 (rather than the original 2.1 to 4.7 ECS range from CMIP5 (Meehl et al., 2020) and the 1.8 to 5.6 ECS range from CMIP6 (Hausfather, 2019)). From Figure S6 it can be seen that ensembles with narrower ECS ranges see a reduction in inter-model spread growth alongst time for the high emission scenarios.

Figure 5 shows the spatial distribution of the projected summer warming by the high emission scenario for CMIP5, CMIP6 and HighResMIP in the three future reference periods. JJA warming is significant and robust for the three future periods in the Mediterranean region (see Fig. 5). HighResMIP warming shows many non-statistically significant grid points, due to the ensemble only having 4 models as it reduces the degrees of freedom for the Student's t-test and makes it harder to reject the hypothesis that the ensemble means for the baseline and the future periods are the same. As seen before, CMIP6 warms more than CMIP5 and at a faster rate. Nevertheless, there is good spatial agreement between the warming projected by the CMIP experiments over the Mediterranean region. The Iberian peninsula, the Balkans and Eastern Europe are the regions with the largest mean summer warming, with values reaching over 8 ℃.

The remaining scenarios also project robust and significant warming for summer throughout the century with a tendency of smaller positive trends by 2050 (not shown). CMIP6 systematically projects higher warming than CMIP5 again with a similar spatial warming pattern. The regions with larger warming are the Iberian peninsula and the Balkans.

The temperature spatial changes during winter for the high emission scenario are shown in Fig. S8. The north-eastern Mediterranean shows the largest projected warming in winter (4.5 ℃ according to CMIP5 and 6 ℃ to CMIP6). For the near-

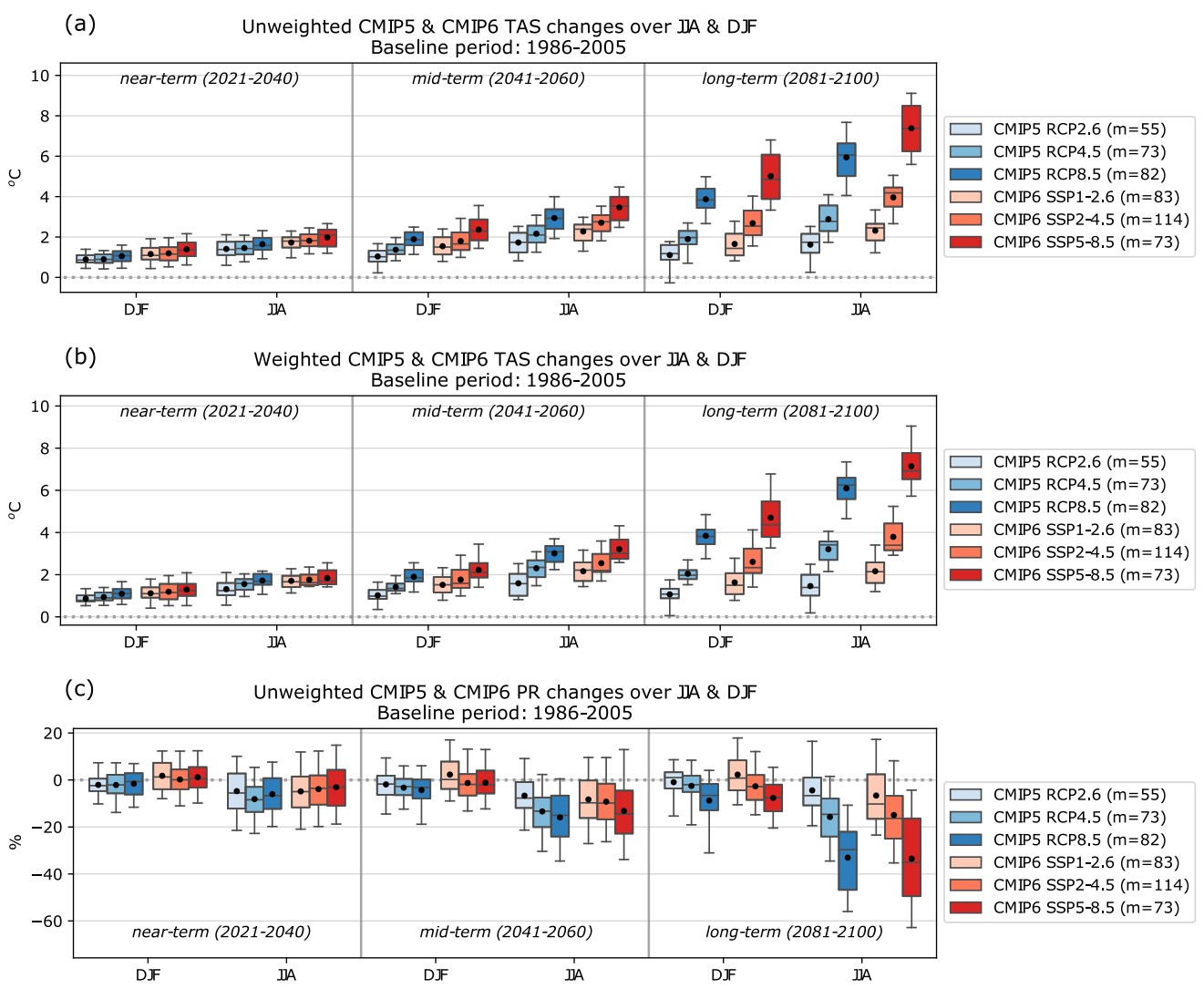

**Figure 4.** CMIP5 and CMIP6 summer and winter projections for the near, mid and long-term periods with respect to the baseline period considering the 2.6, 4.5 and 8.5 $Wm^-2$ RCP and SSP radiative forcing scenarios for (a) unweighted ΔTAS (b) weighted ΔTAS and (c) unweighted ΔPR. The black horizontal line in the boxes represents the median and the black dot is the mean. The interquartile range (IQR) and whiskers are defined by the 25th-75th and 5th-95th percentiles, respectively. The number of members in the boxplot distributions is represented by *m* in the legend.

term, HighResMIP shows a slightly larger TAS increase than CMIP6 in eastern Europe. The rest of scenarios agree with the spatial distribution of changes but with lower warming magnitudes (not shown).

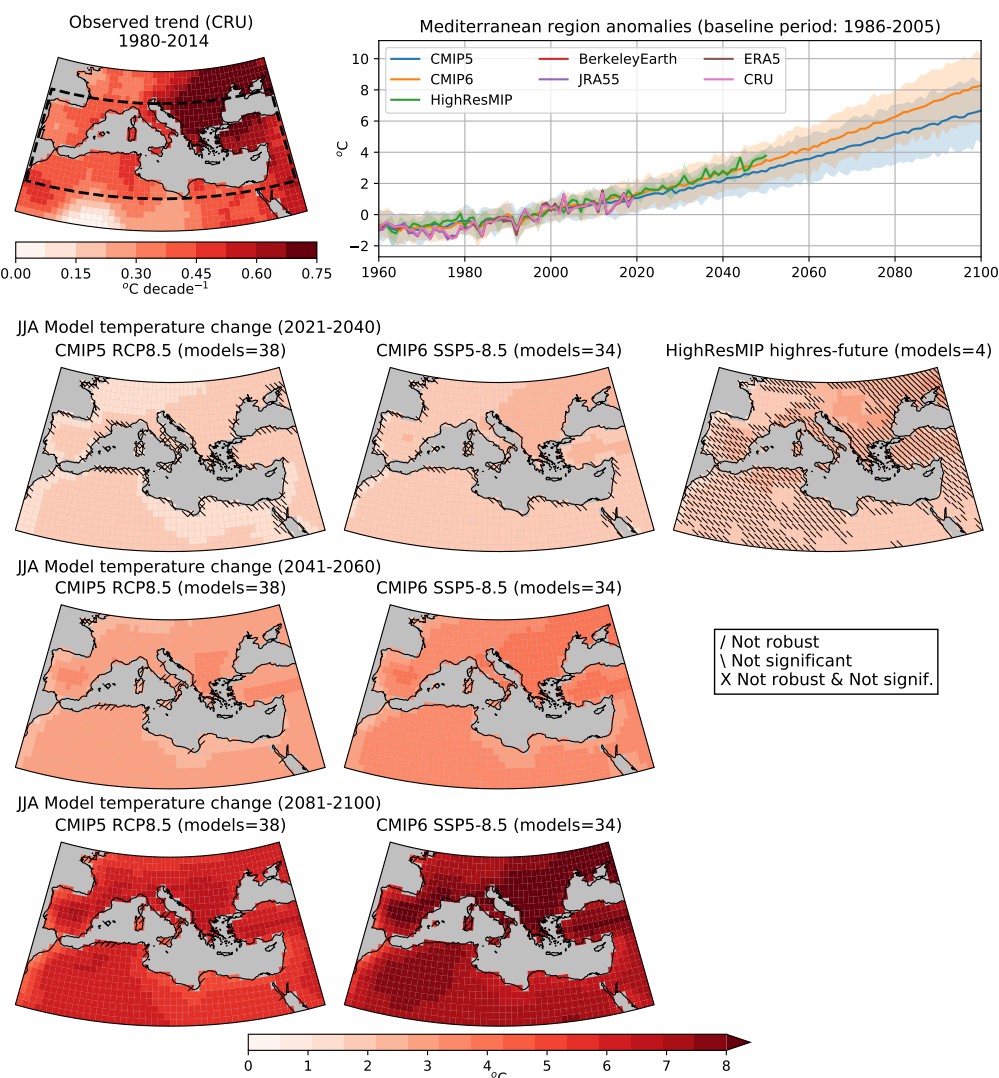

**Figure 5.** Summer ΔTAS according to CMIP5, CMIP6 and HighResMIP ensemble means (columns) for the three relevant future periods (rows), under the RCP8.5 and SSP5-8.5 scenarios. The time series plot shows the anomalies in the Mediterranean region with respect to the period 1986-2005 for the multi-model ensembles and the observational references. A solid line indicates the one-member-per-model ensemble mean and the shaded region indicates the 5th-95th percentiles range. CRU trend for the period 1980-2014 is shown along with the dashed line which bounds the Mediterranean region. Non-significant coastline grid points are due to differences in the original grid resolutions between models. Coarse models have masked data in complex coastline regions once regridded, making the ensemble smaller and therefore reducing the degrees of freedom for the t-Student test.

### 3.3.2 Precipitation

In contrast to temperature, CMIP5 and CMIP6 show the same mean summer $\Delta$PR declines of -33 % by the end of the century under the high emission scenario (Fig. 4.c). CMIP6 has a wider inter-model 90 % range than CMIP5. The former spans from -63 % to -4 % and the latter from -56 % to -11 %. For the low emission scenario CMIP6 mean JJA precipitation declines by -7 % (90 % between -23 % and +17 %) and CMIP5 by -4 % (90 % within -19 % to +16 %). In winter and by the end of the century, CMIP6 precipitation declines by -8 % (90 % between -20 % and +5 %) and CMIP5 by -9 % (90 % between -31 %

to +4 %) under the high emission scenario. For the low emission scenario in DJF, CMIP6 shows a mean +2 % precipitation increase (90 % between -11 % and +18 %) and CMIP5 a -1 % decline (90 % within -15 % to 9 %). The rest of seasons and scenarios show mean $\Delta$PR declines beginning from the mid-term period onwards. Nevertheless, during the 21st century under the low emission scenario a slight increase in mean winter precipitation is projected. HighResMIP near-term projections of PR change are contained within the CMIP6 ensemble (Fig. S5(b,d)). Generally, the signal is considerable, but the inter-model

spread is wide for all multi-model ensembles, therefore we will later present the statistical robustness and significance of changes. Finally, from the area averaged distributions of $\Delta$PR (Fig. 4.c) we can see that the largest source of uncertainty is the forcing scenario for long-term summer projections, and the inter-model spread for winter and near and mid-term summer.

Precipitation spatial changes in the Mediterranean region only get more robust and significant with time (see Fig. 6). $\Delta$PR projected for the long-term during summer, and under the 8.5 $Wm^{-2}$ scenarios, indicate significant and robust decline for most

310 of the region. Note that neither significant nor robust changes are projected in the south and east Mediterranean mainly due to the already low or non-existent precipitation during summer, according to the climatology observed by CRU. Both CMIPs agree on the south-western Iberian peninsula having the strongest precipitation decline, with long-term CMIP6 changes ranging from -50 to -60 % and CMIP5 by -30 to -40 % for the high emission scenario. Despite lower forcing scenarios projecting non-robust and non-significant changes (except the western Mediterranean for long-term SSP2-4.5), the results agree on a general

precipitation decline throughout the region with patterns similar to the high emission scenario projections (not shown). The HighResMIP projections agree with CMIP6 mean magnitudes and spatial pattern for most of the seasons in the near-term period (the large amount of non-robust and non-significant grid points must be noted).

$\Delta$PRs in winter are different from those in summer (see Fig. S9). The southern part of the domain is expected to see a significant and robust precipitation decline in the long-term of up to -20 to -40% over northern Africa. The north of the

320 Mediterranean is located in a transition zone, as precipitation in areas northward from the Pyrenees, Alps and Balkans is projected to increase and in areas under 38º N is projected to decrease, causing changes for the Iberian, Italian and Balkan peninsulas to remain uncertain. In comparison to CMIP5, CMIP6 shows more significant and robust changes over the region and wider 5-95th percentile spreads. This remains true for the rest of the scenarios (not shown).

As a final remark, the observed winter precipitation variability in the time series is stronger than the simulated 90 % inter-

325 model spread (5th-95th percentiles shown as shades in Fig. 6), this suggests that models fail to fully capture the amplitude of precipitation inter-annual oscillations during winter.

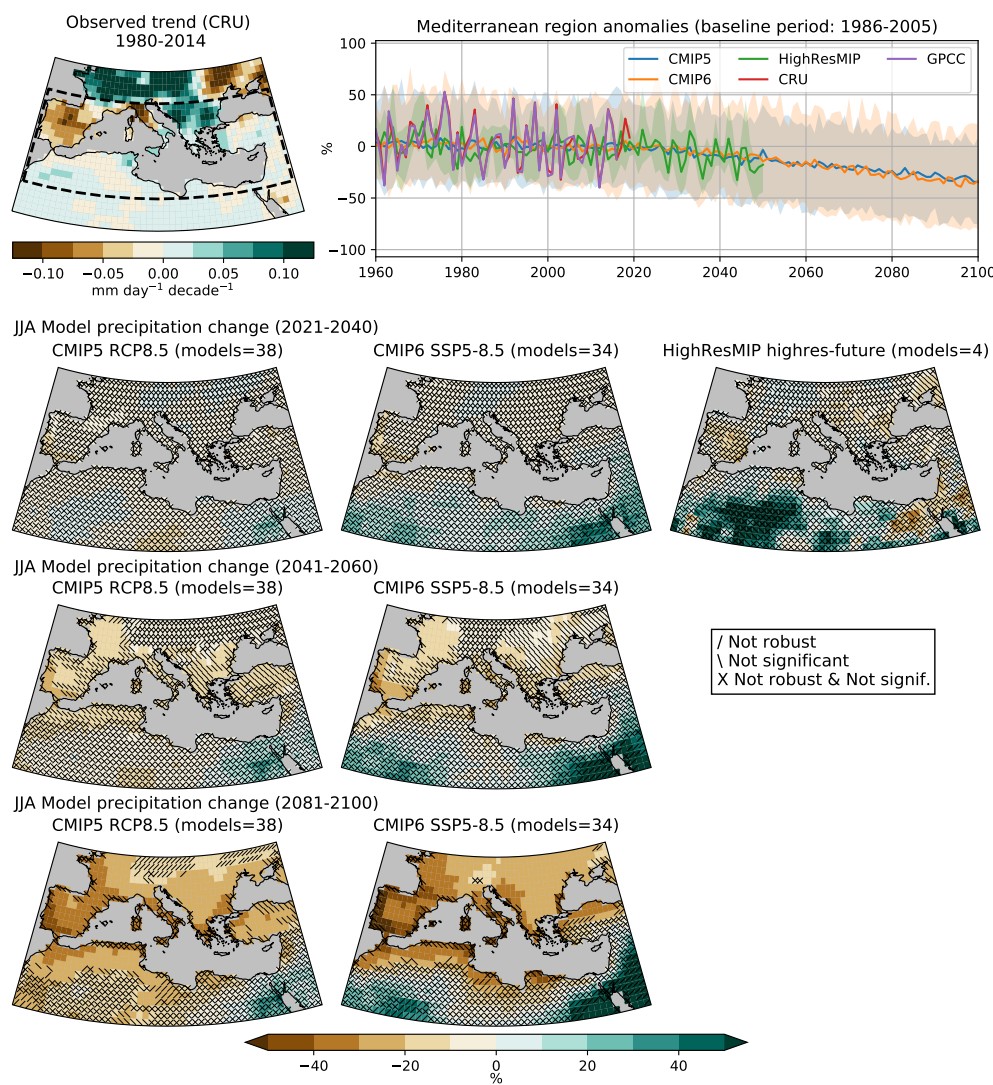

**Figure 6.** Same as Fig. 5 for summer precipitation and showing CRU in the top left panel.

## 3.4 Weighted projections

The models of CMIP ensembles perform very differently depending on the computed diagnostic, and some models share similarities. Section 1 of the supplementary material explains in further detail how differently models represent the observed climate over the Mediterranean region, justifying the need to constraint the projection ensembles.

We obtain new projections from applying the performance and independence weighting method to TAS projections from the CMIP5 and CMIP6 ensembles. Figure 4.b shows the distribution of $\Delta$TAS in the weighted ensembles for the three emission scenarios and the three future periods. The weighting increases the CMIP5 mean and median projections while at the same

time decreasing the CMIP6 mean and median projections, bringing the two ensemble means closer together: before weighting, CMIP5 and CMIP6 medians differed 1.32 ºC and after weighting the difference is 0.68 ºC (for the highest emission scenario in JJA). Generally, the high emission scenario means are the ones that see larger reductions in CMIP6 ensemble, e.g. differences between the unweighted and weighted ensemble means are of around -0.3, -0.2 and -0.1 ºC in summer and winter for SSPs 5-8.5, 2-4.5 and 1-2.6, respectively. The IQRs are generally narrowed for all seasons, and scenarios except for the mid and late century summer SSP2-4.5, SSP1-2.6 and RCP2.6 scenarios. The 90 % spreads are slightly reduced or maintained, exceptions are the CMIP6 DJF long-term distributions and the CMIP6 JJA low and mid emission scenarios for the mid-term. The 75-95th percentile range in the weighted CMIP6 ensemble increases while the 5-25th percentile range decreases, generating a skewed CMIP6 weighted distribution towards smaller warming. Weighting the CMIP5 ensemble leads to a more constrained distribution.

The weighted $\Delta$TAS projections in winter show similar responses as in summer: the mean signal in CMIP6 decreases while it increases in CMIP5, making the differences between both mean distributions smaller. In some cases the weighting did not lead to significant alterations of the projected inter-model spread, suggesting that uncertainties in the temperature changes are well sampled by the original ensembles. In contrast, the large IQR of CMIP6 model projections in the long-term is reduced by half, and the CMIP5 90 % inter-model spread narrows up to 1 ºC, after weighting. Nevertheless, even though the weighting approach reduces the probability of the most extreme warming values, they remain possible in the weighted ensemble. Generally speaking, the 90 % inter-model spreads are maintained while the IQRs narrow.

To assess the contribution of the performance and independence weights in the resulting distribution we have plotted the distribution of performance and full weights, and compared the raw ensemble long-term warming distribution with the performance-weighted and the fully-weighted warmings (Fig. S12). Summer performance shifts both CMIP ensembles to larger warmings, while the addition of independence weights shifts CMIP6 median to lower warmings than the raw ensemble. Winter performance weights don't have an effect on the warming medians but they narrow CMIP5 spread. The addition of winter independence weighting shifts CMIP6 median warming and broadens its inter-model spread. CMIP5 median remains unchanged but its spread grows toward the raw distribution without reaching it.

Note that precipitation weighted projections are not shown as there is no evidence that the diagnostics used to assess temperature (Merrifield et al., 2020) are relevant to the precipitation response of the models.

## 4  Discussion

Projections obtained from climate multi-model ensembles contain various sources of uncertainties. Different modelling methods and emission scenarios (e.g land use, GHG emissions...) lead to different results (Tebaldi and Knutti, 2007). We use different multi-model ensembles and radiative forcing scenarios to consider as many factors as possible contributing to the uncertainty of the Mediterranean climate change projections. Additionally, a weighting method constraining the projections has been applied to reduce uncertainty in the projections.

We have shown that average Mediterranean temperature changes were larger than the global-mean average during summer, but close to it during winter, for all scenarios, time periods and model ensembles. This hotspot is projected to enhance over the 21st century under the scenarios RCP8.5, SSP5-8.5, RCP4.5 and SSP2-4.5, and to diminish from the mid to long term under the RCP2.6 or SSP1-2.6 scenarios. Interestingly, the multi-model ensemble mean projections of the low emission scenario
show a recovery of the precipitation decline towards the end of the century, suggesting that precipitation could be restored to historical values relatively fast in the Mediterranean region if strict mitigation policies are applied. Previous studies also have identified the Mediterranean warming amplification (Lionello and Scarascia, 2018; Zittis et al., 2019), but it must be stressed that this enhanced warming does not apply to the winter season.

We argue that the different results obtained from CMIP5 and CMIP6 on the Mediterranean hotspot and the unweighted
projections are largely due to the global response from each multi-model ensemble. Figs. 3, S2 and S4 show how the regional changes relative to the larger scale are similar for both CMIPs, indicating that CMIP6 isn't producing a regional enhancement of climate change, but it rather follows a larger global change. This behaviour is most evident in summer than in winter, as the relative changes with respect to larger scales are more similar for the two multi-model ensembles. To further support this statement, we can look at the spatial distribution of changes within the mediterranean region in Figs. 5, 6, S3, S8 and S9. The
figures generally agree on the spatial distribution of changes even if the magnitudes differ. Therefore, we can argue that the main difference in TAS and PR output from the older (CMIP5) and newer generation (CMIP6) multi-model ensembles is an enhancement of the global change, while its relation with the Mediterranean region response has been maintained. The work from Palmer et al. (2021) arrives at a similar conclusion for the European region.

The drivers of the projected Mediterranean climate change has been stuided Brogli et al. (2019), Brogli et al. (2018) and
385 Tuel and Eltahir (2020). They have found that the mechanisms projected to drive the Mediterranean climate are large-scale upper-tropospheric flow changes (winter PR), reduction in the regional land-sea temperature gradient (winter and summer PR), meridional Hadley cell shift (summer PR, winter and summer TAS) and changes in the north-south lapse-rate contrast. While these drivers have been deeply studied for CMIP5, affirmirming that the same mechanisms remain valid for the CMIP6 ensemble would be speculative.

Consistent with basic radiative forcing theory (Wallace and Hobbs, 2006), temperature projections have shown that the warming over the 21st century is larger when stronger radiative forcing scenarios are applied. There is confidence in a precipitation decline for the high emission scenario over the whole Mediterranean region in summer and only in the south during winter. Conclusions should be drawn carefully from precipitation as there is a large inter-model spread. For other seasons and scenarios, precipitation declines are projected, although results are uncertain due to large spread and low significance and
robustness over most of the region. Regarding HighResMIP, the HR near-term precipitation and temperature changes generally fall within the CMIP6 ensemble distribution and no clear improvement could be seen from the increased resolution in the historical trends, probably due to the small number of HighResMIP models available for the assessment, and the focus on larger scale changes and temporal resolutions.

The largest source of uncertainty to determine the warming and precipitation change by the mid and long-term periods is
400 the emission scenario (as seen in Figure 4). To illustrate the scenario uncertainty, let's take the range between the 5th and 95th

percentile of the low (high) and high (low) emission scenario distributions for temperature (precipitation) changes. CMIP6 shows a range from 1ºC to 9ºC warming and -62% to 19% precipitation long-term changes in summer. CMIP5 ranges from 0.1ºC to 7.5ºC warming and -54% to 18%. This broad spectrum of possible futures has various possible outcomes associated. The inter-model spread grows at faster rates along the 21st century with higher radiative forcing, in part due to the different climate sensitivities of the models inside the ensemble, i.e. the differences between a low and a high climate sensitivity model will get amplified with larger radiative forcing.

The implications of an 8.5 $Wm^{-2}$ increase in radiative forcing from preindustrial times by the end of the century could pose severe strains on: human health, due to heat-related illness (Lugo-Amador et al., 2004) and altered transmission of infectious diseases (Patz et al., 2005); food security due to crop pests and diseases (Newton et al., 2011) and productivity decline in many countries which economies depend on agriculture (Devereux and Edwards, 2004); and water insecurity due to droughts (Devereux and Edwards, 2004) and changing rainfall patterns in vulnerable regions (Sadoff and Muller, 2009). Note that the three climate change induced impacts defined above are closely intertwined and may increase existing scarcities.

In face of the very pessimistic future projected by the high emission scenario, some studies argue that 8.5 $Wm^{-2}$ forcing is highly unlikely as it is based on an expansion of the coal use along the 21st century instead of on a reduction (Ritchie and Dowlatabadi, 2017a). In the context of energy transition and lowering demand of coal, the high emission scenario is often criticised (Ritchie and Dowlatabadi, 2017b). Nevertheless, studies on the carbon cycle discuss that $CO_2$ feedbacks might be underestimated in the GHG-concentration scenarios (Booth et al., 2017), and thus we've considered keeping the 8.5 scenarios as an extreme but yet possible future.

The CMIP6 ensemble is known to have models with notably higher climate sensitivity than CMIP5, i.e. radiative forcing generates stronger changes and at a faster rate (Hausfather, 2019). Higher sensitivity can be due to model design or the definition of the radiative forcing scenario. Even if SSP and RCP scenarios are labelled after the radiative forcing (in $Wm^{-2}$) by the end of the century, the transient GHG concentrations are different (Meinshausen et al., 2011; Riahi et al., 2016). Wyser et al. (2020) suggests that running the same model with equal 2100 GHG concentrations from SSP and RCP (2.6, 4.5 and 8.5 $Wm^{-2}$), leads to larger temperature changes when forcing the model with the former. It has been argued that improvements in the formulation of clouds and aerosols in CMIP6 are the major contributors to larger climate sensitivities with respect to CMIP5 (Meehl et al., 2020; Hausfather, 2019). Even if there is higher sensitivity to radiative forcing in some CMIP6 models, this behaviour is not reproduced by all of them, resulting in a larger inter-model spread compared to CMIP5.

In terms of which multi-model ensemble performs better, there are some studies that attribute better performance of the CMIP6 ensemble compared to historical references in China (Zhu et al., 2020), Turkey (Bağçaci et al., 2021), the Tibetan plateau (Lun et al., 2021), and the global mean (Fan et al., 2020). Nevertheless, as no performance studies have been made specifically in the Mediterranean region we can not speculate which ensemble performs better. Therefore, it would be a topic of interest for further study.

Assessing the weighted temperature ensemble, we found that the CMIP6 distribution shifts to lower changes, meaning that models showing larger TAS changes have been down-weighted, reducing the differences between CMIP6 and CMIP5 experiment medians and means. To find the reason behind this shift we plotted the ensemble warming distribution for the long

term after applying only the performance weights (numerator of equation 1) and compared it to the raw and fully weighted ensembles (see Fig. S12). We found that the indepence weights are the ones shifting the CMIP6 ensemble to lower warmings rather than the performance. In this regard, CMIP5's median is unaltered by the independence and its effect can only be seen in inter-model spread changes. Summer performance weights shift CMIP5 and CMIP6 to larger warmings, suggesting that a number of the members projecting larger changes do a better job at representing the historical climate. A last remark that can be extracted from Fig. S12 is that both independence and performance weighting play an important role which changes between seasons and ensembles. Therefore, there is not a straightforward interpretation of the general behaviour of the weights.

Precipitation weighted projections are not shown in this study as we have no proof that the diagnostics used to assess temperature are relevant to evaluate the model's precipitation response, and therefore the authors will consider it for further work.

## 5   Conclusions

This study aims to analyse the projected temperature and precipitation changes by the CMIP5 and CMIP6 multi-model ensembles in the Mediterranean region. Different scenarios and seasons have been assessed to tackle the uncertainties inherent to ensemble projections. To complement the traditional information provided, a weighting method that accounts for historical performance and inter-independence of the models has been applied to offer an alternative view of the temperature projections.

The Mediterranean is a climate-change hotspot due to the amplified warming and drying when compared to the large-scale climate behaviour. The amplified warming of the Mediterranean especially affects temperature during summer and not in winter. Comparing the Mediterranean hotspot in CMIP5 and CMIP6 we found that the ratio of warming amplification is similar for both multi-model means, meaning that no enhanced warming is projected by the CMIP6 ensemble, but it is rather the consequence of a globally warmer ensemble.

Conclusions must be drawn carefully from multi-model ensembles as the single models perform very differently and might share dependencies with each other. Model agreement gives high confidence in significant and robust warming affecting the entire Mediterranean region along the 21st century caused by anthropogenic emissions. The Balkans during winter and the Balkan and Iberian peninsulas during summer are expected to be the most affected regions. Precipitation changes are less robust and significant and show greater spatial heterogeneity than the warming. Significant and robust declines in precipitation are expected to affect the Mediterranean in summer and the southern part in winter by the end of the 21st century if high emission scenarios are considered. The warming combined with a precipitation decline could put under strain the whole region, especially the south, which has less resources to adapt to the changing climate. The biggest source of uncertainty to determine the magnitude of TAS and PR changes is the emission scenario, which will depend on the future policies and measures for mitigation followed. Considering three scenarios, the range of the long-term projected warming (given by the 50 % inter-model spread) can go from 1.83-8.49 ℃ according to CMIP6 and 1.22-6.63 ℃ according to CMIP5 in summer. For precipitation, the decline ranges go from -49 to -16 % in CMIP6 and -47 to -22 % in CMIP5. It has also been concluded that part of the increasing warming inter-model spread with time is related to the wide range of ECS values among the ensemble members.

A weighting method has been applied to reduce the uncertainty caused by models that poorly represent key aspects of the historical climate or by the high dependence of the results provided by families of models (that might be overrepresented in the multi-model ensemble). Based on the constrained projections we conclude that CMIP6 overestimates warming in the Mediterranean and its 25th to 50th percentile inter-model spread. The shift to lower warming seen by the CMIP6 weighted ensemble is driven by the independence weighting. CMIP5 slightly underestimates warming and generally overestimates the IQR inter-model spread. The weighted projections are relevant because they help to reconcile the conclusions extracted from the last two CMIP phases, reducing future uncertainties of climate change. The fact that CMIP6's 90 % spread range is unaltered, shows that the climate uncertainty might have been underestimated in previous, less physically advanced, CMIP exercises, which displayed smaller inter-model spread when constrained.

Further work is required for the weighting method to identify the most relevant diagnostics that best assess historical precipitation model performance. As spatial heterogeneities can be seen in the Mediterranean region, we suggest considering subregions for the Mediterranean to extract more user-relevant information from the constrained projections. Furthermore, it would be of great interest for the community to update studies on the physical mechanisms and the performance of the CMIP6 multi-model ensemble in the Mediterranean region.

*Code and data availability.* The tool used for the diagnostics (ESMValTool) can be found at https://github.com/ESMValGroup/. ESMVal-Tool v2.2 is publicly available on Zenodo at https://zenodo.org/record/4562215#.YOWlhTqxVH4 (Andela et al., 2021). The source code of the ESMValCore package, which is installed as a dependency of the ESMValTool v2.3, is also publicly available on Zenodo at https://zenodo.org/record/4947127#.YOa2BsBR1QI (Andela et al., 2021b). ESMValTool and ESMValCore are developed on the GitHub repositories available at https://github.com/ESMValGroup. The observational data used: GPCC (doi:10.5676/DWD_GPCC/FD_M_V2020_025), CRU (https://crudata.uea.ac.uk/cru/data/hrg/cru_ts_4.04/cruts.2004151855.v4.04/, https://doi.org/10.1038/s41597-020-0453-3.), JRA55 (https://jra.kishou.go.jp/JRA-55/index_en.html#reanalysis), ERA5 (https://doi.org/10.1002/qj.3803), BerkeleyEarth (http://berkeleyearth.lbl.gov/auto/Global/Gridded/Complete_TAVG_LatLong1.nc), HadSLP (https://doi.org/10.1175/JCLI3937.1). CMIP data: all the CMIP5 and 6 datasets were downloaded from the Earth System Grid Federation (ESGF). The models used are listed in Appendix A. For CMIP6, the DOIs of the datasets from the ESGF can be obtained in Identifier DOI after clicking on "show citation" from the following url https://esg-dn1.nsc.liu.se/search/cmip6-liu/?source_id=ACCESS-CM2,ACCESS-ESM1-5,AWI-CM-1-1-MR,BCC-CSM2-MR,CAMS-CSM1-0,CAS-ESM2-0,CESM2-WACCM,CIESM,CMCC-CM2-SR5,CNRM-CM6-1,CNRM-ESM2-1,CanESM5-CanOE,EC-Earth3,FGOALS-f3-L,FGOALS-g3,FIO-ESM-2-0,GFDL-ESM4,GISS-E2-1-G,HadGEM3-GC31-LL,INM-CM4-8,INM-CM5-0,IPSL-CM6A-LR,KACE-1-0-G,KIOST-ESM,MCM-UA-1-0,MIROC-ES2H,MPI-ESM1-2-HR,MPI-ESM1-2-LR,MRI-ESM2-0,NorESM2-LM,NorESM2-MMUKESM1-0-LL&experiment_id=historical,ssp126,ssp245,ssp585&variant_label=r1i1p1f1,r1i1p1f2,r1i1p1f3,r2i1p1f1,r2i1p1f2,r2i1p1f3,r3i1p1f1,r3i1p1f2,r4i1p1f1,r4i1p1f2,r5i1p1f1,r5i1p1f2,r6i1p1f1,r6i1p1f2,r7i1p1f1,r7i1p1f2,r8i1p1f1,r8i1p1f2,r9i1p1f1,r9i1p1f2&table_id=Amon&variable_id=tas

The ESMValTool recipes and the code for the diagnostics can be found at http://doi.org/10.23728/b2share.01b483fa953241b2b2d8f5242cae6e8c

Additional figures not shown in the main text or the supplementary material can be found in the figure repository built with a shiny app following the link https://earth.bsc.es/shiny/medprojections-shiny_app/.

## Appendix A: Model data summary

A summary of all the initial-condition runs from the multi-model ensembles CMIP5, CMIP6 and HighResMIP, for the three radiative scenarios used in this study can be found in Table A1.

**Table A1.** Summary of the members used in this study from CMIP5, CMIP6 and HighResMIP. The columns display the emssion scenarios.

| CMIP5 | lat° $x$ lon° | RCP2.6 | RCP4.5 | RCP8.5 | CMIP6 | lat° $x$ lon° | SSP1-2.6 | SSP2-4.5 | SSP5-8.5 |
|---|---|---|---|---|---|---|---|---|---|
| ACCESS1-0 | 1.25°$x$1.875° | - | r1i1p1 | r1i1p1 | ACCESS-CM2 | 1.25°$x$1.875° | r1i1p1f1 | r(1-2)i1p1f1 | r1i1p1f1 |
| ACCESS1-3 | 1.25°$x$1.875° | - | r1i1p1 | r1i1p1 | ACCESS-ESM1-5 | 1.25°$x$1.875° | r(1-3)i1p1f1 | r(1-10)i1p1f1 | r(1-3)i1p1f1 |
| BCC-CSM1-1 | 2.8125°$x$2.8125° | r1i1p1 | r1i1p1 | r1i1p1 | AWI-CM-1-1-MR | 0.9375°$x$0.9375° | r1i1p1f1 | r1i1p1f1 | r1i1p1f1 |
| BCC-CSM1-1-M | 1.125°$x$1.125° | r1i1p1 | r1i1p1 | r1i1p1 | BCC-CSM2-MR | 1.125°$x$1.125° | r1i1p1f1 | r1i1p1f1 | r1i1p1f1 |
| BNU-ESM | 2.8125°$x$2.8125° | r1i1p1 | r1i1p1 | r1i1p1 | CanESM5 | 2.8125°$x$2.8125° | r(1-10)i1p1f1 | r(1-10)i1p1f1 | r(1-10)i1p1f1 |
| CanESM2 | 2.8125°$x$2.8125° | r(1-5)i1p1 | r(1-5)i1p1 | r(1-5)i1p1 | CanESM5-CanOE | 2.8125°$x$2.8125° | r(1-3)i1p1f1 | r(1-3)i1p1f1 | r(1-3)i1p1f1 |
| CCSM4 | 0.942406°$x$1.25° | r(1-5)i1p1 | r(1-5)i1p1 | r(1-5)i1p1 | CAS-ESM2-0 | 1.40625°$x$1.40625° | - | r(1,3)i1p1f1 | - |
| CESM1-BGC | 0.942406°$x$1.25° | - | r1i1p1 | r1i1p1 | CESM2 | 0.9375°$x$1.25° | r1i1p1f1 | r(1,4,10-11)i1p1f1 | r(1,2)i1p1f1 |
| CESM1-CAM5 | 0.942406°$x$1.25° | r(1-3)i1p1 | r(1-3)i1p1 | r(1-3)i1p1 | CESM2-WACCM | 0.9375°$x$1.25° | r1i1p1f1 | r(1-3)i1p1f1 | r1i1p1f1 |
| CMCC-CESM | 3.75°$x$3.75° | - | - | r1i1p1 | CIESM | 0.9375°$x$1.25° | - | r1i1p1f1 | r1i1p1f1 |
| CMCC-CM | 0.75°$x$0.75° | - | r1i1p1 | r1i1p1 | CMCC-CM2-SR5 | 0.9375°$x$1.25° | r1i1p1f1 | r1i1p1f1 | r1i1p1f1 |
| CMCC-CMS | 1.875°$x$1.875° | - | r1i1p1 | r1i1p1 | CNRM-CM6-1 | 1.40625°$x$1.40625° | r(1-6)i1p1f2 | r(1-6)i1p1f2 | r1i1p1f2 |
| CNRM-CM5 | 1.40625°$x$1.40625° | r1i1p1 | - | r(1-2,4,6,10)i1p1 | CNRM-CM6-1-HR | 0.5°$x$0.5° | r1i1p1f2 | r1i1p1f2 | r1i1p1f2 |
| CSIRO-Mk3-6-0 | 1.875°$x$1.875° | r(1-10)i1p1 | r(1-10)i1p1 | r(1-10)i1p1 | CNRM-ESM2-1 | 1.40625°$x$1.40625° | r(1-5)i1p1f2 | r(1-5)i1p1f2 | r1i1p1f2 |
| EC-Earth | 1.125°$x$1.125° | r(8,12)i1p1 | r(2,6-9,12-14)i1p1 | r(1,2,6,8,9,12,13)i1p1 | EC-Earth3 | 0.703125°$x$0.703125° | r(4,6,9,11,13,15)i1p1f1 | r(2,7,18-24)i1p1f2 | r(4,6,9,11,13,15)i1p1f1 |
| FGOALS-s2 | 1.6667°$x$2.8125° | - | r1i1p1 | r(1-3)i1p1 | FGOALS-g3 | 2.25°$x$2° | r1i1p1f1 | r(1-4)i1p1f1 | r1i1p1f1 |
| FIO-ESM | 2.8125°$x$2.8125° | r(1:3)i1p1 | r(1-3)i1p1 | r(1-3)i1p1 | FGOALS-f3-L | 1.0°$x$1.25° | r1i1p1f1 | r1i1p1f1 | r1i1p1f1 |
| GFDL-CM3 | 2.0°$x$2.5° | - | r1i1p1 | r1i1p1 | FIO-ESM-2-0 | 0.942408°$x$1.25° | r(1-3)i1p1f1 | r(1-3)i1p1f1 | r(1-3)i1p1f1 |
| GFDL-ESM2G | 2.0°$x$2.5° | r1i1p1 | - | r1i1p1 | GFDL-ESM4 | 1.0°$x$1.25° | r1i1p1f1 | r1i1p1f1 | r1i1p1f1 |
| GFDL-ESM2M | 2.0°$x$2.5° | r1i1p1 | - | r1i1p1 | GISS-E2-1-G | 2.0°$x$2.5° | r1i1p3f1 | - | r1i1p3f1 |
| GISS-E2-H | 2.0°$x$2.5° | r1i1p1 | r(1-3,5)i1p1 | r(1-2)i1p1 | HadGEM3-GC31-LL | 1.25°$x$1.875° | r1i1p1f3 | r1i1p1f3 | r(1-3)i1p1f3 |
| GISS-E2-H-CC | 2.0°$x$2.5° | - | - | r1i1p1 | INM-CM4-8 | 1.5°$x$2.0° | r1i1p1f1 | r1i1p1f1 | r1i1p1f1 |
| GISS-E2-R | 2.0°$x$2.5° | r1i1p1 | r(2,6)1i1p3 | r(1-2)i1p1 | INM-CM5-0 | 1.5°$x$2.0° | r1i1p1f1 | r1i1p1f1 | r1i1p1f1 |
| GISS-E2-R-CC | 2.0°$x$2.5° | - | - | r1i1p1 | IPSL-CM6A-LR | 1.25°$x$2.5° | r(1-4,6)i1p1f1 | r(1-6,10,11,14,22,25)i1p1f1 | r1i1p1f1 |
| HadGEM2-AO | 1.25°$x$1.875° | r1i1p1 | - | r1i1p1 | KACE-1-0-G | 1.25°$x$1.875° | r(1-2)i1p1f1 | r(1,3)i1p1f1 | r1i1p1f1 |
| HadGEM2-ES | 1.25°$x$1.875° | r(1-4)i1p1 | r(1-4)i1p1 | r(1-4)i1p1 | KIOST-ESM | 1.875°$x$1.875° | - | r1i1p1f1 | - |
| INMCM4 | 1.5°$x$2.0° | - | r1i1p1 | r1i1p1 | MCM-UA-1-0 | 2.25°$x$3.75° | r1i1p1f1 | r1i1p1f1 | r1i1p1f1 |
| IPSL-CM5A-LR | 1.875°$x$3.75° | r(1-4)i1p1 | r3i1p1 | r(1-4)i1p1 | MIROC6 | 1.40625°$x$1.40625° | r(1-3)i1p1f1 | r(1-3)i1p1f1 | r(1-3)i1p1f1 |
| IPSL-CM5A-MR | 1.26761°$x$2.5° | r1i1p1 | r1i1p1 | r1i1p1 | MIROC-ES2L | 2.8125°$x$2.8125° | r1i1p1f2 | r1i1p1f2 | r1i1p1f2 |
| IPSL-CM5B-LR | 1.875°$x$3.75° | - | r1i1p1 | r1i1p1 | MPI-ESM1-2-HR | 0.9375°$x$0.9375° | r1i1p1f1 | r1i1p1f1 | r1i1p1f1 |
| MIROC-ESM | 2.8125°$x$2.8125° | r1i1p1 | r1i1p1 | r1i1p1 | MPI-ESM1-2-LR | 1.875°$x$1.875° | r(1-10)i1p1f1 | r(1-10)i1p1f1 | r(1-10)i1p1f1 |
| MIROC-ESM-CHEM | 2.8125°$x$2.8125° | r1i1p1 | r1i1p1 | r1i1p1 | MRI-ESM2-0 | 1.125°$x$1.125° | r1i1p1f1 | r1i1p1f1 | r1i1p1f1 |
| MIROC5 | 1.40625°$x$1.40625° | r(2-3)i1p1 | r(2-3)i1p1 | r(2-3)i1p1 | NESM3 | 1.875°$x$1.875° | r(1-2)i1p1f1 | r(1-2)i1p1f1 | r(1-2)i1p1f1 |
| MPI-ESM-LR | 1.875°$x$1.875° | r(1-3)i1p1 | r(1-3)i1p1 | r(1-3)i1p1 | NorESM2-LM | 1.25°$x$3.75° | r1i1p1f1 | r(1-3)i1p1f1 | r1i1p1f1 |
| MPI-ESM-MR | 1.875°$x$1.875° | r1i1p1 | r(1-3)i1p1 | r1i1p1 | NorESM2-MM | 0.9375°$x$1.25° | r1i1p1f1 | r1i1p1f1 | r1i1p1f1 |
| MRI-CGCM3 | 1.125°$x$1.125° | r1i1p1 | r1i1p1 | r1i1p1 | UKESM1-0-LL | 1.25°$x$1.875° | r(1-4,8)i1p1f2 | r(1-4,8)i1p1f2 | r(1-4,8)i1p1f2 |
| MRI-ESM1 | 1.125°$x$1.125° | - | - | r1i1p1 | | | | | |
| NorESM1-M | 1.875°$x$2.5° | r1i1p1 | r1i1p1 | r1i1p1 | | | | | |

| HighResMIP | lat° $x$ lon° | SSP5-8.5 | | lat° $x$ lon° | SSP5-8.5 | | lat° $x$ lon° | SSP5-8.5 |
|---|---|---|---|---|---|---|---|---|
| CMCC-CM2-HR4 | 0.942406°$x$1.25° | r1i1p1f1 | CNRM-CM6-1-HR | 0.5°$x$0.5° | r1i1p1f1 | HadGEMGE3-GC31-HM | 0.234375°$x$0.351562° | r1i1p1f1 |
| CMCC-CM2-VHR4 | 0.234681°$x$0.3125° | r1i1p1f1 | EC-Earth3P | 0.703125°$x$0.703125° | r3i1p2f1 | HadGEMGE3-GC31-MM | 0.555557°$x$0.833333° | r1i1p1f1 |
| CNRM-CM6-1 | 1.40625$x$1.40625 | r1i1p1f1 | EC-Earth3P-HR | 0.3515625$x$0.3515625 | r2i1p2f1 | | | |

## Appendix B: Diagnostics, $\sigma_d$ and $\sigma_s$ of the weighting method

This Appendix aims to describe the methodology behind the performance and independence weighting. First, we will explain the diagnostics chosen to compute the distances and secondly how to obtain the two constant shape parameters from equation (1).

As the aim is to obtain weighted projections from a multi-model ensemble, the diagnostics to assess performance and independence must be relevant for the used variable. The weighting is going to be optimised for temperature projections and therefore variables TAS and PSL from the historical period (1980-2014) will be used, as these variables are relevant for the projected temperature ((Merrifield et al., 2020), (Brunner et al., 2020)). In order for CMIP5 to comply with the historical reference period, the diagnostics will include the first years of the scenario experiments (2006-2014). As there is a unique ensemble of members for each project, scenario and season, each ensemble will have its own set of weights.

The diagnostics used are differences, climatologies, trends, and variability. According to Tebaldi and Knutti (2007), TAS historical trends have an evident physical link and high correlation with future projected warming. The trend is defined by the linear ordinary least square regression fit for each grid point with time as independent variable during the reference period (TREND); the climatologies are computed as the time mean of each grid point over the reference period (CLIM); the differences are computed by subtracting the area averaged climatology to each grid point's reference period climatology (DIFF) and the variability is obtained with the mean inter-annual standard deviation for each grid point (STD). As the trend is not relevant for PSL, it is not computed (Merrifield et al., 2020).

When assessing performance, the aim is to identify the models that more faithfully represent the historical climate. As all our results are computed as differences from the historical period, model biases in the climatology shouldn't be relevant. That is why the diagnostics used for performance weighting are TAS-TREND, TAS-DIFF, TAS-STD, PSL-DIFF and PSL-DIFF. Differently, the aim of weighting for independence is to identify members that have similar traits. Biases in models should be similar for dependent models (Merrifield et al., 2020), therefore we use CLIM for temperature and sea level pressure (TAS-CLIM and PSL-CLIM) to compute the distances $S_{ij}$ from equation (1). Computing the climatology over relatively long periods is a good approach as the internal variability gets minimised and ideally, it is the main attribute distinguishing two members of the same model (Hawkins and Sutton, 2011).

Finally, to compute the actual values of $D_i$ and $S_{ij}$ the single diagnostic distances (e.g. TAS-TREND, TAS-DIFF, PSL-DIFF...) must be combined. It is done by normalizing the single diagnostics with the median over all members and then averaging them.

The shape parameters are constant thresholds that inform how large or small distances should be to determine performance ($D_i$) and independence ($S_{ij}$). If $\sigma_d$ is overconstrained (small value), it will generate a very strict performance weighting as only members with very low values of $D_i$ will receive any weight. Contrarily, if high values of $\sigma_d$ are used, models with large distances will receive performance weight, leading to too permissive constraints. The independence shape parameter doesn't work in such a straightforward way, small values of $\sigma_s$ could weight all models as being independent, as the distance to consider two members dependent would have to be too small. This could result in models receiving similar weights. A similar thing could happen but for the opposite reason if a large $\sigma_s$ was used i.e. most models would seem dependent as large distances between members would be considered small enough. We therefore must find an optimal $\sigma_s$ that is neither too small nor too large (Knutti et al., 2017).

The ensemble gives the necessary information to make a best guess of both shape parameters. Regarding the performance parameter, Knutti et al. (2017) suggests applying perfect model tests for a range of $\sigma_d$ candidates to obtain the optimal mag-

nitude. The candidates are values between the 10% and 200% of the median $D_i$ distance. Consecutively, all members in the ensemble are once taken as the reference while the rest are weighted following equation (1) with $D_i$ being the distance between the perfect member and the member $i$. The $\sigma_d$ candidates are iteratively tested for all perfect model tests until the smallest $\sigma_d$ that makes 80% of the perfect models fall in between the 10th and 90th percentiles of their respective weighted ensembles is found. The diagnostics used in the test are the same as the ones used to weight performance but computed for the future periods (2041-2060 and 2081-2100) as we want $\sigma_d$ to be based on the uncertainties of the future projection ensemble. The average $\sigma_d$ between both periods is used for its corresponding season, scenario and CMIP ensemble.

The parameter $\sigma_s$ is informed by models with more than one initial-condition run. Ideally, members from the same model should be considered completely dependent as their modelling assumptions are the same, even though internal variability makes the runs differ. The independence weighting should identify when initial-condition runs from the same model are added or subtracted from an ensemble. If the independence weights (equation (1) denominator) are calculated for an ensemble with one member per model ($w_j^{ind}$) and then all the available members of a model $j$ are added to the ensemble ($E_j$ represents the amount of members added), the average independence weights of model $j$ ($\tilde{w}_j^{ind}$) are expected to decrease by a ratio $1 : E_j$. Additionally, including members of a model $j$ to the ensemble should have a minimal effect on the independence weights of the rest of models $i$ represented by only one member in the ensemble.

The optimal $\sigma_s$ is found via an iterative process for a range of $\sigma_s$ candidates, looking for the one that minimizes the sum $\epsilon_1 + \epsilon_2$, where $\epsilon_1$ and $\epsilon_2$ are defined as (Brunner et al., 2019):

$$mean_j \left[ w_j^{ind}(\sigma_s) + E_j - \tilde{w}_j^{ind}(\sigma_s) \right]^2 = \epsilon_1$$
$$mean_j \left\{ mean_i \left[ w_{i \neq j}^{ind}(\sigma_s) - \tilde{w}_{i \neq j}^{ind}(\sigma_s) \right]^2 \right\} = \epsilon_2 \; \forall j$$

*Author contributions.* JC, FD and MJ designed the study. JC developed and ran the diagnostics, and wrote the initial manuscrip. MJ helped in the Figures production. MJ, FD, RM and JC contributed to the interpretation of the results and improvement of the manuscript. PB and MS contributed to the download and fixes of the datasets used in this study.

*Competing interests.* The authors declare that they have no conflict of interest.

*Acknowledgements.* We acknowledge the World Climate Research Programme, which, through its Working Group on Coupled Modelling, coordinated and promoted CMIP5 and CMIP6. We thank the climate modeling groups for producing and making available their model output, the Earth System Grid Federation (ESGF) for archiving the data and providing access, and the multiple funding agencies who support CMIP5

and CMIP6 and ESGF. We also thank the European Center for Medium-Range Weather Forecast (ECMWF), the Japan Meteorological Agency (JMA), the University of East Anglia (UEA), the Deutscher Wetterdienst (DWD), Berkeley Earth and the Met Office (UKMO) for
providing ERA5, JRA55, CRU, GPCC, BerkeleyEarth and HadSLP2 reanalysis/observational data, respectively. We acknowledge the Earth System Evaluation Tool (ESMValTool) community for the development and distribution of the tool, and we sincerely thank the technical support from Saskia Loosveldt-Tomas and Javier Vegas-Regidor (BSC-CNS) with the tool. The work in this paper was partly supported by the European Commission H2020 project EUCP (grant no. 776613).

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
