# Peer review of "The Mediterranean climate change hotspot in the CMIP5 and CMIP6 projections"

_Earth System Dynamics, 2021_

## Author Comment (AC3)

Supplementary material from answer to RC3

[Figure]

(a)

(b)

**Figure 1:** Scatter plot displaying the performance weights (left) and the full weights (right) against their 2081-2100 warmings with respect to 1986-2005 for DJF (a) and JJA (b). The boxplots represent the long-term warming distributions for weighted (darker colour) and unweighted (lighter colour) from the CMIP5 (blue) and CMIP6 (red) ensembles.

[Figure]

**Figure 2:** Scatter of the single-member relation between performance weight (left) and the full weight (right) against the model's ECS for DJF (a) and JJA (b). The least-squares linear regression fits are represented by the blue (CMIP5) and red (CMIP6) lines.

---

## Author Response (AR1)

**Remarks to follow the point-by-point answers:**

**1) comments from referees/public, (2) author's response, and (3) author's changes in the manuscript.**
**1., 2., 3., … = comments from referees**
**1) = author's response**
**2) author's changes in the manuscript**

**The line numbers refer to the revised version. NOT to the track-changes file.**
* * *
**point-by-point reply to #R1 comments:**

*1.* ***Abstract.*** *The abstract is too general. You need to be more specific and give detailed numerical results of the study (the same comment also applies to the conclusion – no numerical values…) In the current version, no numerical values (temperature or precipitation changes, with uncertainties) are given even though they are an important outcome of this study (especially the CMIP5/CMIP6 comparison, and the model weighting). Please consider modifying the abstract (and conclusion) to include them (e.g. ll. 11-12 summer precipitation change, ll. 9-10 how much more warming in CMIP6 vs. CMIP5, etc.) Also, some of the wording is quite general, like l. 14 "in some regions" > please specify.*

   1) The authors agree that the abstract seems too general and lacks numerical results of the study
   2) Numerical results of the study have been added to the revised version. The specific suggestions in ll. 9-14 have been noted and modified in the abstract (ll. 10-18).

*2.* ***Methods section.*** *I find it a bit difficult to read. Most of the elements are there but the structure could be improved. For instance, you begin with "all computations were performed with…" But what computations do you perform? They are only introduced later at l. 122. It would be clearer to begin by saying what computations you perform, and then specify all the technical details (regridding, land vs. ocean, etc.) Maybe also consider having a separate sub-section for the weighing approach.*

   1) The authors agree that the Methodology section is hard to read as all the different metrics and diagnostics are presented together.
   2) The section has been restructured. We have divided it into Projections verification (l.121), Mediterranean hotspot evaluation (l.129), Mediterranean projected changes quantification (l.142) and Weighting method (l.160)

*3.* ***Weights definition.*** *The paragraph on the calculation of weights is not very clear. How exactly are Di and Sij are calculated? Is Di = sqrt((TAS-DIFF)\*\*2+(TAS-STD)\*\*2+(TAS-TREND)\*\*2+(PSL-DIFF)\*\*2+(PSL-STD)\*\*2)? Appendix B is not very helpful either as it does not contain the equation. Similarly, the definition of Sij is unclear. Showing an equation would be much better. Additionally, in one sentence the diagnostics are said to be "the 20-year PSL and TAS climatologies" and in the*

*next you say that the diagnostics are computed over the 35-year period. Two different time periods are also used for Di and Sij, which is confusing (see also my next comment) Also, what observational reference do you choose (in DIFF)? The mean calculated over all observation/reanalysis products?*

1) We always use the 35-year period in the diagnostics used to compute Sij and Di. The 20-year periods are used only when showing the projected results. Therefore, the two periods aren't mixed within the weighting method. What might have caused some confusion is that the periods 2041-2060 and 2081-2100 are used when finding the magnitude of the optimal shape parameter σd (we need to check how the multi-model ensemble reacts to different values of σd to assess if it would cause and under or over-constraint of the future projections).

2) Equations showing how to calculate Di have been added to clarify the methodology (ll.191-192). Regarding time periods, Table 2 (l.199) has been added to clarify which diagnostics use which periods.

*4.      **Baseline periods.** The fact that you use two different periods is confusing. First, 20 years is a bit short to calculate averages. 30 years is usually preferred. You mention that 20 years of data are heavily influence by inter-annual variability; that is true for trends, but for averages also. The extra 10 years of observations should also be used to assess GCM performance. Since you have to merge the historical and RCP8.5 simulations in CMIP5 to calculate trends for the 1980-2014 period, why not merge them to calculate averages also? The issue with having those two reference periods is that you mix them in the calculation of weights, which is not very consistent.*

1) The fact that we used 20 year periods to show mean changes is consistent with the work conducted in the IPCC AR6. As answered in the previous point the weighting method, the verification of the models against observations and the display of the future projections are independent things. For the weights we use the 35-year historical period to ensure that the trend is well represented, the same is true for the verification with observations and finally, when showing the future projected changes we follow the IPCC guidelines to display changes against the baseline period.

2) This has been clarified in the text with the addition of Table 2. And justifying it with the IPCC 2021 reference (l.145).

*5.      **Trend significance.** It seems that to detect statistical significance the authors are implementing a t-test to determine whether the ensemble-mean average trend (in TAS or PR) is significantly different from zero. But that is not really appropriate. Trend statistical significance should be assessed for each model separately based on its inter-annual variability. The spread in trend values across models is not related to the magnitude of the trends themselves. For instance, there can be a large spread across models (+1,+2,+5,+10°C) but each trend may be statistically significant for the corresponding model (because inter-annual variability is smaller for the +1°C model than for the +10°C model). A better definition of significance in this context might be the fraction of models for which the trend is significant (or, like robustness, to impose that the trend is significant for at least 80% of models) This could change the conclusions for HighResMIP*

1) While this is an interesting approach to compute the statistical significance of the results, we consider that it wouldn't correspond to the significance of the results as we show them. The results are obtained using the change in the multi-model ensemble mean. To evaluate if the changes are significant we need information about the difference in ensemble distribution between the historical and future periods, not the significance in the changes of single models. Therefore, we compare the ensemble-mean and model spread between the historical and future periods.

2) The authors haven't modified the text in this regard (see justification above).

6. **Results.** The section is a bit difficult to read. Maybe you could try to structure it a bit more? For instance, in section 3.2 you move from temperature to precipitation back to temperature and precipitation again, switching between scenarios, seasons and periods.

1) The authors agree that jumping back and forth from temperature to precipitation results can make the results section difficult to read.

2) The revised text has the temperature (l.259) and precipitation (l.293) separated into subsubsections, hopefully making the text easier to follow.

7. ***Discussion.*** *Despite the emphasis on the "hotspot" aspect, the discussion contains no information on the physical mechanisms responsible for the existence of the Mediterranean hotspot. Some literature exists on the topic (e.g., Brogli et al. https://doi.org/10.1175/JCLI-D-18-0431.1, Tuel et al. https://doi.org/10.1175/JCLI-D-20-0429.1) Please consider adding a short discussion on the comparison of the hotspot between CMIP5 and CMIP6, and the links to the known/likely physical mechanisms.*

1) The authors will consider adding information about the mechanisms that drive the hotspot in the revised manuscript's discussion.

2) In the discussion section, we have added references to publications that explore the physical mechanisms of climate change in the Mediterranean. As the current literature has only used CMIP5, we don't want to be speculative and, therefore, we don't assume that the same mechanisms remain true for CMIP6. It is a nice addition to the discussion nevertheless.

**Technical comments:**

*l.1 "increased warming trend" -> maybe "enhanced warming trend"?*

2) Suggestion added in the text (l.1).

*l.1 "makes" -> "make"*

2) Suggestion added in the text (l.1).

*l.2 "historical and scenario" -> missing "future"?*

2) Suggestion added in the text (l.2).

*l.4 "following scenarios RCP2.6, SSP1-2.6, RCP4.5, SSP2-4.5, RCP8.5 and SSP5-8.5" -> I suggest separating the CMIP5 and the CMIP6 scenarios*

2)  Suggestion added in the text (l.4).

*l.7 "along" -> "over" (and "over" on l. 8 can become "across" or "during")*

2)  Suggestion added in the text (l.8).

*l.9 "being CMIP6" -> "CMIP6 being"*

2)  Suggestion added in the text (l.9).

*l.17 "continental" -> "oceanic" as well. A continental climate is not "humid and mild".*

2)  Suggestion added in the text (l.21).

*ll.17-19 This sentence is unclear. Can you please rephrase?*

2)  The sentence has been rephrased. (ll.21-23)

*l.24 "global warming mean" -> "global-mean warming"*

2)  Suggestion added in the text (l.28).

*l.26 add "are" before "projected"*

2)  Suggestion added in the text (l.30).

*ll.26-28 Unclear what this sentence refers to here…*

2)  The sentence seems to lack context, therefore it has been removed.

*l.37 "tools" -> you mean GCMs?*

2)  Suggestion added in the text (l.40).

*l.59 "assumption" -> "criteria"?*

2)  Suggestion added in the text (l.62).

*l.62 "presented in section 3"*

2)  Suggestion added in the text (l.65).

*l.69 It is not just PSL that is used to calculate model weights; TAS is used also, correct?*

2)  It has been clarified that TAS is also used (l.74).

*l.79 "mangnitudes"*

2)  Suggestion added in the text (l.82).

*l.82 "has" -> "have"*

2) Suggestion added in the text (l.85).

*l.89 "initial conditions" (no "-")*

2) Suggestion added in the text (ll.94-95).

*l.96 "containing" -> "including"*

2) Suggestion added in the text (l.101).

*l.106 "differences in the thermodynamic properties of the surfaces" -> "differences in surface thermodynamic properties"*

2) Suggestion added in the text (l.119).

*Figure 1 -> it would be useful to have also the numerical values for the global/latitudinal mean changes*

1) While it is true that some information could be gathered from the global/latitudinal mean changes, the figure is already very crowded and, from the point of view of the authors, adding 12 magnitudes could distract the reader from the main message of the image.
2) The suggestion hasn't been implemented as argued above.

*Table 1. -> Please specify in the caption the variables corresponding to the acronyms (TAS, PR, etc.), or specify later at l.104 when the variables are introduced.*

2) The acronyms are already defined in (l.72), nevertheless, they have also been added to Table 1 caption for the convenience of readers.

*l.145 Did you introduce variable M?*

2) M has been changed to m and it is defined in l.172.

*l.149 "weight" -> "more weight"?*

2) Suggestion added in the text (l.178).

*l.167 "30-45N latitudinal belt mean" -> Why not all land regions? One could argue that to make it a global hotspot one should compare against all other land areas (say of the same size). One issue also is that both the Mediterranean and the 30-45N belt contain many grid points with very small precipitation averages -> potentially large relative changes which may bias the analysis.*

*Also, you compare to 30-45N values but only over land, right? In that case Figure S3 should not have data over the oceans. For the sake of readers who are not used to land-only values ("global" often means land and ocean), I suggest you specify "global land mean", e.g., at l.166.*

1) We tried several options to illustrate the precipitation hotspot. We compared the Mediterranean precipitation to the global precipitation and the precipitation in the

30-45ºN latitudinal belt to reach the same conclusion: the Mediterranean is projected to experience larger changes than the global mean and the regions with the same latitude. This already stands out when looking at the global maps of precipitation change. Using land-only points in the latitudinal belt does not change the conclusions because many land regions in these latitudes experience important precipitation increases (e.g., South Asia).

2) The methodology to extract the results from Figures S2 and S4 have been reworked to avoid computing the area-averaged relative changes (explained in ll.140-141), which could lead to potentially large/small values that could bias the analysis. The code in the B2SHARE repository has been updated.

*l.176 "projects larger precipitation increases in regions where the hotspot has a negative sign such as the southeast of the domain" -> unclear. Larger increases where the change is negative?*

1) A negative sign hotspot refers to grid points where the precipitation increases with respect to the latitudinal band change. The authors agree that the term is a bit confusing.
2) To avoid confusions and maintain the point that we wanted to convey, we have removed "where the hotspot has a negative sign" and left "the southeast of the domain" (l.228).

*l.179 "larger scale means" -> "global average"*

2) Suggestion added in the text (l.230).

*Figure S5: What are OBS? It would be better to show here the values for the different observation/reanalysis products. Or at least their mean and the range across products (maybe that is what is currently shown, and if yes, please specify in the caption) In HighResMIP values the different markers are also a bit too small to see the difference. Make them bigger maybe?*

1) In Figure 1, OBS shows the values for the different observation/reanalysis products.
2) It has been specified in Figure 1's caption that the OBS distribution is formed of the different observational products results.

*l.206 "for the remaining seasons" -> "for MAM and SON" (or specify in the previous sentence that you look at DJF and JJA).*

2) Suggestion added in the text (l.267).

*l.210 "trend" -> "trends"*

2) Suggestion added in the text (l.209).

*l.211 "but the PR high-resolution (HR) models trends display outliers in summer" -> "but some of the high-resolution (HR) models exhibit trends outside the CMIP6 range for PR in summer"*

2) Suggestion added in the text (ll.209-210).

*Figures 3 and S6: Could you please add horizontal grid lines? Right now it is difficult to look at this figure and see the differences between weighted and unweighted results.*

    2) The figures have been modified to add horizontal lines. Additionally, panels *a* and *b* now specify that the results are unweighted.

*l.220 "under for" -> "under"*

    2) Suggestion added in the text (l.264).

*l.227 "cannot be drawn". Still, you could compare the HighResMIP values with those of the corresponding, low-resolution climate model versions.*

1) The exercise of comparing the HR with LR results has been done and no general behaviour can be seen. In part due to the small sample size.
2) The authors have added a sentence explaining that no conclusive behaviour between HR and LR models can be seen (ll.269-270)

*l.229 delete "respectively"*

    *2)* Suggestion added in the text (l.294).

*l.237 Figure S5, not S3*

    *2)* Suggestion added in the text (l.303).

*ll.237-238 "Generally, the signal is weak and the inter-model spread is wide for all multi-model ensembles" -> what does this refer to? Precipitation projections only? If yes "weak" is not really appropriate. Mid-to-long term trends in JJA precipitation are large (-15% or below)*

    *2)* Sentence removed from the main text.

*ll.240-244 What is the conclusion here? If you constrain model ECS then you will get a smaller spread in projections.*

1) Exactly, this is the conclusion, which is shown later in section 5. (ll.465-466)
2) The sentence has been rephrased to better introduce what is concluded in section 5. (ll.276-277)

*l.248 "Student's t-test"*

    *2)* Suggestion added in the text (l.281).

*l.253 "CMIP6 systematically projects" instead of "keeps projecting"*

    *2)* Suggestion added in the text (l.287).

*l.260 "precipitation changes only get more robust and significant with time" -> does this mean that temperature changes don't get more robust and significant with time?*

1) The sentence does not imply that the temperature is not getting more robust and significant with time. Temperature changes are generally robust and significant from the start, therefore precipitation differs from temperature in the sense that it only gets robust and significant in some regions for the more distant future.
2) The sentence hasn't been modified following the justification above.

*l.265-267 Please rephrase.*

*2)* It has been rephrased (l.314).

*l.267 "concord" -> "agree"*

2) Suggestion added in the text (l.315).

*l.272-273 It sounds like you are saying that precipitation both increases and declines in the Balkans.*

1) The first part of the sentence refers to the region at the north of the Balkans and the second to the Balkan peninsula.
2) The text has been modified to make the distinction more explicit. (ll.319-321)

*l.276-277 This sentence comes a bit out of nowhere. Also, what is the 90% range? Please clarify.*

1) The sentence tries to highlight the fact that the observed winter variability is not matched by the inter-model spread of both CMIPs
2) The sentence has been rephrased to make it clearer. (ll.324-325)

*l.278 suggest "Weighted projections" to be consistent with section 3.2*

2) Suggestion added in the text (l.326).

*l.293 "The mean signal in CMIP6 decrease whereas it increases in CMIP5"*

2) Suggestion added in the text (l.344).

*ll.297-298 "Nevertheless, even if the probability of a future extreme-warming decreases, such temperature increases are still considered valid by the weighted ensemble" -> I suggest rephrasing along the lines of "Nevertheless, even though the weighting approach reduces the probability of the most extreme warming values, they remain possible in the weighted ensemble".*

2) Suggestion added in the text (l.348).

*l.304 "Mediterranean"*

2) Fixed in the text (l.363).

*ll.306-307 suggest rephrasing as "We have shown that average Mediterranean temperature changes were larger than the global-mean average during summer, but close to it during winter, for all scenarios, time periods and model ensembles."*

   2) The sentence has been rephrased (ll.365-366).

*l.324 "no clear improvement could be seen from the increased resolution" -> did you compare the HighResMIP models with the lower-resolution versions of the same GCMs?*

   1) The authors compared both resolutions and no improvement or general behaviour could be seen. partly due to the small size of the sample.
   2) The text remains the same.

*l.330 "The largest source of uncertainty to determine the warming and precipitation change by the mid and long-term periods is the emission scenario." Where did you show that? Is it true for both TAS and PR?*

   2) This is shown in the results, but it is gathered/discussed just after the sentence in l. 397.

*l.365 "Precipitation weighted projections are not shown in this study as we have no proof that the diagnostics used to assess temperature are relevant to evaluate the models' precipitation response." -> you could still weigh models based on their past precipitation trends, no?*

   2) From the author's knowledge, diagnostics to properly constrain the precipitation are yet to be defined. The diagnostics need to be physically and statistically consistent with the models' dynamics and they need to be independent between them. Further work needs to be conducted to find the adequate diagnostics for PR.
* * *
**point-by-point reply to #R2:**

*1.     Regarding precipitation: The eastern Mediterranean is characterized by almost completely dry summers and precipitation mostly during winter. Therefore, including it in JJA in the precipitation calculation over the whole Mediterranean basin seems to me problematic. This is seen in the verification against observations for (Fig. S5), and in the lack of robustness and significance in changes (Figs.4). I strongly suggest that the authors do their calculations (not only verification, but all the calculations) considering the very significant differences in precipitation between different regions over the Mediterranean basin.*

   1) From our point of view, this shouldn't be an issue as the area aggregations we have applied are computed with the absolute rather than the relative values. Nevertheless, this was not the case for the MedHS scatter plots. Therefore the scatter plots need to be re-done.

2) The scatter plots in Figures 1, S2 and S4 have been re-done using absolute values in the area aggregations

*2.     Model data and observational data: A table with a list of the grid sizes of each data set is required.*

1) The authors will consider this suggestion
2) The resolutions have been added in Table A1

*3.     S5, at least the comparison to observations, should be part of the article body and not a supplement. This is part of the heart of the paper: how to quantify the veracity of the simulations and the results seen in this figure are strongly correlated to the remark #1.*

1) We agree that the comparison to observations should be displayed in the body of the article. We will move the historical trend comparison to the main text.
2) We have added the historical trend panels in the main text and a verification subsection has been included in the results (Verification, l.202).

*4.     Due to the complexity of the precipitation analysis I would also suggest limiting the manuscript to temperature only, without the need of additional calculations for precipitation as in remark #1. For instance, as stated in lines 365-366 "Precipitation weighted projections are not shown in this study as we have no proof that the diagnostics used to assess temperature are relevant to evaluate the models precipitation response." Diagnostics should turn clearer by dividing the Mediterranean basin following precipitation climatic characteristics in the different seasons.*

1) From the authors' understanding of the hotspot, it is important to include the effect of precipitation. Regarding the weighting method and precipitation, it is an issue that is yet to be resolved as diagnostics used to constrain temperature are inadequate for precipitation. We plan to further investigate this issue in the future.

   The authors think that splitting the Mediterranean domain escapes the aim of assessing the region as a whole and as defined by the IPCC. Going back to our answer to point 1, the area-averaged relative changes have been aggregated from the single grid-points absolute values, and therefore, results shouldn't be affected by large relative changes in arid regions.

2) We have maintained the precipitation in our work as justified above.

**Minor remarks:**

*Abstract: The sentence: " Results obtained from the model weighting scheme indicate increases in CMIP5 and reductions in CMIP6 warming trends, thereby reducing the distance between both multi-model ensembles." it is not clear what the reference is to the written increases and reductions, and what variable(s) are the authors referring to.*

1) The increases and reductions refer to warming trends as indicated in the sentence.
2) The sentence has been slightly modified to better convey the message.

*38: "while running the same model multiple times under the same experiment samples internal variability (Hawkins and Sutton, 2011)"- using different initial conditions?*

1) This sentence refers to the different members of the same model. Which is the model ran multiple times from the start
2) We have added a clarification in l.42.

*55: "to global-mean and large-scale changes" – Not clear*

1) The authors have rephrased this sentence to make it clearer (ll.57-58).

*86-88: "The results from CMIP5 and CMIP6 sharing the same 2100 radiative forcing will be displayed together for simplicity, but the reader should always bear in mind that the evolution of GHG concentrations differs between them". – Not clear.*

    2) An additional sentence has been added to clarify the fact that the GHG concentration differs for RCPs and SSPs that share the same forcing at the end of the century (ll.91-93).

*108-113: "The baseline periods 1986-2005 and 1980-2014 are the reference to assess the models performance against observations. The shorter 1986-2005 period (from Collins et al. (2013)) serves as a baseline for the calculation of climate change signals. The longer reference period (35 years) is used to compute historical trends, as 20-year trends are considered to be too heavily influenced by internal variability (Merrifield et al., 2020; Peña-Angulo et al., 2020). The reason for using the older 1986- 2005 20-year period instead of the more recent 1995-2014 (Brunner et al., 2020) is to avoid CMIP5's historical period ending in 2005 to overlap with the corresponding scenario projection runs that start in 2006"*

*Not clear. For the historical comparison between CMIP5 and CMIP6 to observations to be statistically consistent to say something about the skill of CMIP5 versus CMIP6 the same period has to be used. From the text and from the periods that these projects are available it seems to me that the comparison against observations is done for different periods. Then we cannot compare between CMIP5 and CMIP6 skills. The text above is not clear. Should it be written separately that 1986-2005 is used for signals and 1980-2014 for performance against observations?*

1) The authors agree with this comment, and it is in line with some issues raised by Referee #1. The text will need to be corrected to be more easily understood.
2) The Methods subsection has been split into multiple subsubsections making it easier to attribute the different time periods to the different diagnostics. With the addition of Table 2, the authors aim to make it easier for the reader to interpret which periods are used for which diagnostics.

*117-119: "The height differences between the model orography and the evaluation grid implies that TAS must be corrected (by means of the 6.49 K/km standard lapse rate) whenever absolute climatologies are used (Weedon et al., 2011; Dennis, 2014)." Not clear*

    2) The sentence has been rephrased (ll.108-109). The authors hope it is clearer now.

*128-129: "A climate change signal is considered robust when at least 80 % of the models agree on the sign of change (Collins et al., 2013)." – Not conditioned to model-observations agreement during the historical period of the aforementioned models?*

1) The authors have considered robustness as defined in Collins et al., 2013, which only takes into account the inter-model agreement on the sign of the projected signal. Adding a condition that depends on the model-observations agreement could be considered a performance metric. When displaying the raw projections we don't aim at considering performance, but rather the behaviour of the ensemble.
2) The robustness methodology for unweighted projections has been maintained.

*158-159: "Each multi-model ensemble" – I understood there is one multi-model ensemble, maybe it should be written each "model ensemble"? Or maybe each "member in the multi-model ensemble?*

1) CMIP5 and CMIP6 are two separate multi-model ensembles. What gets the weights are all the members within this ensemble. We wrote "multi-model" for convenience, but the authors agree that it can be misleading as what gets the weights are the members rather than the models.
2) To avoid confusion we have removed "multi-model" and added "(CMIP5 and CMIP6) in l.196.

*165-168: "Figure 1 compares warming differences of the high radiative forcing scenarios of CMIP5 and CMIP6 over the Mediterranean with respect to the 1986-2005 global mean for winter, summer and the annual means. For precipitation, Mediterranean change is compared to the 30° N-45° N latitudinal belt mean. The Mediterranean region shows a higher annual temperature increase than the global mean." – As written here it is very unclear. What was compared here? My guess is as follows: (a) temperature and precipitation change was calculated globally against 1986-2005, (b) the same as (a) was done only for the Mediterranean region, (c) in the plots we see (b) minus (a). Is this correct? This is not written above.*

1) The authors agree that this sentence does not communicate properly what the hotspot figure shows.
2) The sentence has been changed so that the description of the figure is adequate (ll.216-217).

*Fig. 1: Either I missed it or there is no discussion at all about the positive change in precipitation shown in JJA panels of CMIP5 RCP8.5 and CMIP6 SSP5-8.*

2) It has been noted in line 228.

*206: "Mostly, for the remaining seasons,…"- What is not the "remaining season"? It seems a sentence is missing before*

2) The text has been modified to explicitly state which seasons we refer to (l.205).
* * *
**point-by-point reply to #R3:**

*1.	The manuscript would be of higher value if there are some comparisons with similar works performed in other geographic sectors of the world. For example, there are some recent efforts focusing on regional climate issues in China (Zhu et al. 2020, 2021, Li et al. 2021: https://doi.org/10.1007/s00376-020-9289-1 ; https://doi.org/10.1016/j.scib.2021.07.026 ; https://doi.org/10.1007/s13351-021-0067-5).*

1) The authors will consider adding such a comparison in the revised manuscript discussion.
2) Lines 494-498 have been added to compare the current work with comparisons between the performance of CMIP5 and CMIP6. No conclusions in the Mediterranean can be drawn in terms of which ensemble performs better. Therefore, we have identified it as further work.

*2.	Although CMIP5's RCP scenarios are close to CMIP6's SSP scenarios with the relevant nomenclature, there are indeed subtle differences for greenhouse gases, especially for emission of aerosols. This seems ignored in the present manuscript. In a more general manner, differences between CMIP5 and CMIP6, as analysed in the manuscript, include many aspects involving both anthropogenic emissions and improvement of models' physics and resolution. It seems that one cannot make a clear idea or conclusion, with what presented in the manuscript.*

1) There is no clear statement that can be drawn without being speculative in terms of CMIP5 and CMIP6 differences. Therefore, the authors have based their understanding of the differences between CMIPs on the current literature, such as differences in the cloud feedbacks, aerosol forcings and aerosol-cloud interactions.
2) With the new iterations of Figures S2 and S4 the authors have been able to explain some similarities between CMIPs Mediterranean temperature change and the amplified precipitation hotspot of CMIP5. It is discussed in ll.432-444

*3.	The ensemble-processing algorithm, based on models' performance and independence, imposes an observation constraint. The authors state that its use can make closer the results of CMIP5 and CMIP6, and make smaller the spreading of each ensemble among its members. They also point out a few exceptions. Are there any explanations? Generally speaking, the manuscript seems a little too descriptive and lacks physical interpretation.*

1) The authors agree that further physical interpretation of the outcome of the weights should be conducted.

   In the manuscript, we based the justification of the CMIP6 ensemble shifting towards CMIP5 on the emergent constraints work from Nijesse et al. 2020 and Tokarska et al.

2020. Their work states that models with higher climate sensitivity aren't consistent with the observed global warming trends. We have conducted further tests and found out that this is not a good explanation for our constraining weights obtained in the Mediterranean region. This is displayed in the figures shown below. Figure 1 represents the performance weights (left) and the full weights (right) against their 2081-2100 warmings with respect to 1986-2005 for DJF (a) and JJA (b). It aims to highlight which effect has each weight (independence and performance) on the ensemble distribution shifts.

The differences between the left and right panels highlight how the independence weighting is the one reducing the warming from the CMIP6 ensemble rather than the performance weighting. CMIP6 Performance-only weights shift the ensemble to the high end of changes for JJA while they keep the ensemble quite unchanged for DJF. Contrarily, the addition of independence weighting shifts it to the lower end for both seasons. It can also be seen that the addition of independence weighting doesn't affect the CMIP5 mean state.

Therefore, an interpretation of the weights that shall be included in the revised manuscript is that CMIP6's distribution is moving towards the CMIP5 ensemble because of the independence weighting effect.

As a final justification of why the simple explanation on the basis of the ECS is not applicable, we display, in Figure 2, the performance (left) and full (right) weights against the model's ECS for DJF (a) and JJA (b). The figure shows how performance gives weight to models with high ECS in JJA and doesn't down-weight them in DJF. Therefore we can't see a clear relationship between low performance and high model's ECS.

In the revised manuscript we will base the justification of the weighted CMIP6 shift on the causes and effects of independence discrimination.

2) The aforementioned figure (Supplementary figure S12) and conclusions have been added to the revised text.

4.  *It is a little disappointing to see only mean climate (for both surface air temperature and precipitation) is processed here, without consideration of any extreme climate events or their representative indices.*

1) The authors agree that it would be very interesting to assess extreme climate events. Nevertheless, we have considered that the paper is already too dense to add this kind of indices. Thanks for the suggestion, it will definitely be part of our future work.
2) As indicated above the authors haven't implemented extreme climate events evaluation.

*5.      Line 165, Figure 1. The figure legend and associated descriptions are confusing for me. "…with respect to the 1986-2005 GLOBAL mean…"; "with respect to the MEAN GLOBAL temperature change and the MEAN 30° N-45° N LATITUDINAL BELT precipitation change". The authors need to clearly indicate what are particular in the displayed graphics, compared to usual practices.*

    1) The authors agree that the titles in Figure 1 are confusing as they say mean global and mean latitudinal belt, while it should be global mean and latitudinal belt mean.
    2) The titles in Figure 1 have been changed

References:

Nijsse, F. J. M. M., Cox, P. M., and Williamson, M. S.: Emergent constraints on transient climate response (TCR) and equilibrium climate sensitivity (ECS) from historical warming in CMIP5 and CMIP6 models, Earth Syst. Dynam., 11, 737–750, https://doi.org/10.5194/esd-11-737-2020, 2020.

B. Tokarska, M. B. Stolpe, S. Sippel, E. M. Fischer, C. J. Smith, F. Lehner, R. Knutti, Past warming trend constrains future warming in CMIP6 models. Sci. Adv. 6, eaaz9549 (2020).

---

## Author Response (AR2)

**R#1 answer to comments:**

Firstly, we would like to thank the thorough comments from Referee #1 as we believe they have added a lot of value to the manuscript. The following is a point-by-point answer to its comments:

*The authors have achieved a thorough revision of their manuscript which is now in my opinion suitable for publication after minor revisions. Below are a few minor details.*

*The only main comment I have concerns the definition of significance for the multi-model mean change (related to one of my initial comments). I remain unconvinced by the statistical soundness of the authors' approach. The t-test for the difference in mean between two samples requires that each sample shares the same distribution, which for the future scenarios is clearly not the case. Inter-annual variability and temperature sensitivity to GHG forcing are different from model to model. Consequently, it doesn't make sense to define a significance level for the multi-model change as is done here. Instead, it should be evaluated on a model-by-model basis (and pooling the results together by — for instance — requiring a minimum number of models exhibiting significance). Don't get me wrong, it still makes sense to compute the multi-model average, of course, but that is not the same as attributing a degree of statistical significance to its change. This remains however a minor point of the paper and it does not modify the conclusions, so I would be fine with leaving it as is, though I think the authors should add a small qualification around ll. 157-158 to clarify the points I raised above.*

Thanks for clarifying the issue you see in the test as we have applied it. To be more consistent in our significance definition, we have clarified the assumptions made to use the t-test as applied in our work. The assumption is that the variances from both reference and future period distributions are equa. Nevertheless, as you mentioned, it should not change the conclusions of the manuscript.

Changed:
A change in the multi-model mean is considered significant when it is beyond the threshold of a two-tailed t-Student test at the 95 % confidence level. The historical and future ensemble mean change and their inter-model standard deviations are used to compute the t-statistic.
to:
A change in the multi-model mean is considered significant when it is beyond the threshold of a two-tailed t-Student test at the 95 % confidence level. We consider that the null hypothesis is met when there is no difference between the multi-model distribution in the reference and future periods, presuming that the variability is stationary and present and future distributions are similar. To compute the t-statistic, first, each model's mean is computed from its members, and secondly, the multi-model ensemble mean and standard deviation are calculated.

*Small comments*

*l. 6 maybe "in the multi-model ensembles" instead of "of the multi-model ensembles"*

Suggestion applied.

*ll. 14-15 "While there is less disagreement in projected precipitation between CMIP5 and CMIP6" -> you mean less disagreement than for temperature, correct? Maybe consider reformulating to make it explicit.*

It has been specified that we are talking about less disagreement with respect to temperature.

*ll. 124-125 "to calculate each of the model's and observational dataset's trends" -> "to calculate trends in each model and observational dataset"*

Suggestion applied.

*l. 125 "dependant" -> "dependent"*

Suggestion applied.

*l. 136 "the whole ensemble differences" -> maybe "differences for each ensemble member"*

Thanks to this comment an error in the description of the Mediterranean hotspot has been found and solved (l.13*,l.136).

*ll. 204-206 "The spread of the multi-model ensemble trends contain the observational ensemble trends (see Fig. 1). Mostly, for seasons SON and MAM, the observations fall inside the 90 % spread of the multi-model ensemble historical runs (not shown)." -> this is not extremely clear, because in the first sentence you do not specify that you are talking about DJF and JJA only. I also don't understand why SON and MAM are different; for DJF and JJA (Fig. 1) observed trends also fall within the multi-model 90% range. You could reformulate in a single sentence, like "PR and TAS trends in the observational ensemble fall within the range of the multi-model ensemble in all seasons (see Fig. 1 for DJF and JJA results)".*

The suggestion has been applied, additionally specifying that results from MAM and SON aren't shown.

*l. 217 "winter, summer" -> it might be better to stick to season acronyms throughout the paper (DJF, JJA, etc.) Right now, you keep jumping from summer/winter to JJA/DJF and vice-versa. (just a suggestion though)*

The authors agree it's better to stick to either summer/winter or JJA/DJF. We have switched to acronyms after defining them in the methods section.

*l. 253 "The results from this figure" -> I would repeat "Fig. S4" for clarity.*

Suggestion applied.

*l. 256 "the low emission scenario panels" -> referenced figure?*

Labels have been added to the figure and are now referenced in the text after mentioning the low emission scenario panels.

*l. 271 "find" -> "found"*

Suggestion applied.

*l. 301 "The rest of seasons" -> "MAM and SON"?*

Suggestion applied.

*l. 302 "Nevertheless, during the 21st century under the low emission scenario a slight increase in mean winter precipitation is projected" -> in MAM and SON? Please specify.*

Suggestion applied.

*l. 324 "the observed winter precipitation variability in the time series" -> "the inter-annual variability in observed precipitation time series" Also, this statement is a bit confusing: how do you see that inter-annual variability in models is lower? Did you check for each individual model?*

This is an unsupported inference we made from the inter-model spread. Therefore, the authors have decided to erase the last part of the sentence.

*l. 383 You might also take a look at Boberg and Christensen (2012) for the CMIP5 case https://doi.org/10.1029/2012GL053650*

Thanks for sharing this work with us. While it is a very interesting publication, the authors consider it is out of the scope of the discussion about how the regional-global projections in CMIP6 are not amplified compared to CMIP5. Even if we haven't implemented the methodology to correct the temperature-dependent biases, we recognise it can be a relevant in-depth study to be considered in the future.

*l. 384 "stuided" -> "studied by"*

Suggestion applied.

*l. 387 Brogli et al. (2018) argued that "a poleward expansion of the Hadley cell is of minor importance for the Mediterranean [temperature] amplification".*

You are right. It was a typo as the text "(summer PR, winter and summer TAS)" should go after the lapse-rate argument, and the Hadley cell shouldn't be mentioned. The text has been corrected.

*l. 443-445 This statement is a repetition from the end of part 3. You could leave it out here.*

Suggestion applied.

*l. 452 "The amplified warming of the Mediterranean especially affects temperature during summer and not in winter" -> "The amplified warming of the Mediterranean is found in summer and not in winter."*

Suggestion applied.

*l. 454 "no enhanced warming" -> specify "no enhanced regional warming" since at the global scale, CMIP6 is indeed warmer*

The authors agree on the suggestion.

**R#2 answer to comments:**

The authors would like to thank Reviewer #2 for all the suggestions made throughout this revision, which helped improve the quality of our work.

*The authors have done a very good job revising the manuscript. Yet, some minor revisions are needed:*
*Lines 43-45: You still have to state that the different ensemble members of a given model result from perturbations to initial conditions? Perturbations to model parameterizations? Running the model several times for the same scenario will lead to an ensemble of realizations only if some changes exist in the different members. Please, look at the experiments design of CMIP5 and CMIP6 and shortly write what perturbations were done.*

The members differ in their initial co at the beginning of the historical run, either by starting from different years in the control run or by introducing perturbations in the last control run timestep. We haven't entered in such detail in the revised text but we have added that members are obtained from differences in the initial conditions (l.43)

---

## Author Response (AR3)

Dear Dr Messori,

The authors have followed your recommendations on the statistical significance assumptions. We have checked if the variability of the distributions is similar and found out that, in most cases, this is not met (especially for the long-term projections). Therefore we have decided to change the test from the "independent samples t-test" to the "paired t-test" (or dependent samples t-test), which allows testing two distributions with differing variabilities. With this new statistic, a slightly wider area of the Mediterranean region has statistically significant precipitation changes, and the low significance of the coastline temperature changes no longer exists in the results.
Overall, the new figures implied changes in a few sentences in the results section, but the discussion and conclusions remain the same. I hope this modification has made our work more robust and is now susceptible to being published in Earth System Dynamics.

Best Regards,
Josep Cos